# Nanofiber Carriers of Therapeutic Load: Current Trends

**DOI:** 10.3390/ijms23158581

**Published:** 2022-08-02

**Authors:** Ivana Jarak, Inês Silva, Cátia Domingues, Ana Isabel Santos, Francisco Veiga, Ana Figueiras

**Affiliations:** 1Laboratory of Drug Development and Technologies, Faculty of Pharmacy, University of Coimbra, 3000-548 Coimbra, Portugal; jarak.ivana@gmail.com (I.J.); nesilva00@gmail.com (I.S.); cdomingues@ff.uc.pt (C.D.); anaicfsantos@gmail.com (A.I.S.); fveiga@ci.uc.pt (F.V.); 2REQUIMTE/LAQV, Group of Pharmaceutical Technology, University of Coimbra, 3000-548 Coimbra, Portugal; 3Institute for Clinical and Biomedical Research (iCBR) Area of Environment Genetics and Oncobiology (CIMAGO), Faculty of Medicine, University of Coimbra, 3000-548 Coimbra, Portugal

**Keywords:** nanotechnology, nanofibers, drug loading, biomedicine, therapeutic applications

## Abstract

The fast advancement in nanotechnology has prompted the improvement of numerous methods for the creation of various nanoscale composites of which nanofibers have gotten extensive consideration. Nanofibers are polymeric/composite fibers which have a nanoscale diameter. They vary in porous structure and have an extensive area. Material choice is of crucial importance for the assembly of nanofibers and their function as efficient drug and biomedicine carriers. A broad scope of active pharmaceutical ingredients can be incorporated within the nanofibers or bound to their surface. The ability to deliver small molecular drugs such as antibiotics or anticancer medications, proteins, peptides, cells, DNA and RNAs has led to the biomedical application in disease therapy and tissue engineering. Although nanofibers have shown incredible potential for drug and biomedicine applications, there are still difficulties which should be resolved before they can be utilized in clinical practice. This review intends to give an outline of the recent advances in nanofibers, contemplating the preparation methods, the therapeutic loading and release and the various therapeutic applications.

## 1. Introduction

The fast progress in the area of nanotechnology has led to new and versatile methods for the fabrication of various nanoscale materials, of which nanofibers (NF) have been in the spotlight because of the variety in the manufacture innovations and applications [1,2]. Nanofibers are defined as nanomaterials with at least one dimension of 100 nm or less, where the length can exceed diameter by 100-times. However, broadly, all fibers with diameter below one μm are considered nanofibers. Unique combination of building materials and fabrication techniques results in nanomaterials of large surface area and surface-to-volume ratio and tunable mechanical and morphological properties which can be exploited for various purposes in the fields related to water and air filtration, automotive and transportation, textiles, medical and biosensor appliances, electronics and energy storage. The nanofiber market has exhibited continuous growth in recent years and is projected to increase significantly in the coming years. The current value is $785 million, and it is expected to reach the compound annual growth rate of more than 15% between 2022 and 2027 [3]. Although filtration devices dominate the market, the demands of the medical and pharmaceutical industries are to be the main fields driving the market growth in the long term. Adjustable surface topology, porosity and pore size can be combined with chemical plasticity to produce materials that can be used for delivery of drugs and biomedicines, and for design of biomedical devices applicable for wound dressing and tissue engineering [4]. Although there are intensive R&D efforts towards nanofiber translation, scale-up of fabrication methods is usually the bottle neck of the process [5]. While several clinical trials of NF formulations have concluded or are currently recruiting or undergoing, there is a limited number of NF products in clinical use [6,7]. Currently, nanofiber-based products used in medicine field involve filtering devices, surgical implants and wound dressings and patches developed for drug delivery [5]. Recent randomized clinical trial (60 patients) concerning the use of acyclovir NF patch for topical treatment of recurrent herpes labialis concluded that although NF patch accelerates symptom relief it is ineffective in shortening healing time [8].

Nanofibers can be manufactured from a variety of natural polymers which include chitosan, fibronectin, gelatin, collagen and silk as well as from artificial polymers such as poly lactic acid (PLA), poly glycolic acid (PGA) and poly lactic-co-glycolic acid (PLGA). Often, polymer combinations or different additives are used to finely tune mechanical and drug loading/releasing properties of resulting NFs. Along the nature of polymer and therapeutic agent, the use of appropriate production method is another parameter in the fabrication of NF-based therapeutic systems and various parameters of these methods need to be optimized on a case-to-case basis. Ideally, NFs used for medical and pharmaceutical purposes should be biodegradable and biocompatible, and should not compromise functional properties of therapeutic load which is particularly applicable in the case of biomedicines such as peptides, proteins, nucleic acids and living cells.

Despite of great promises and advances achieved in the field of NF-based drug and biomedicine delivery vehicles, some unresolved issues still persist and limit their translational potential. This review plans to give an outline of the recent advances in therapeutic nanofibers related to preparation methods, the drug/biomedicine loading and release, and the various therapeutic applications.

## 2. Methods of Preparation

The area of therapeutic NF production is an extremely fertile filed of nanomedicine. Despite of variety of methods that have been developed so far (Figure 1), during the last decade or so, some methods have become more represented in scientific research. The broad applicability in regard to polymer material and therapeutic molecules, controllable morphologies and therapeutic load release, or simplicity of use and ease of scalability are some of the factors that made those methods highly attractive from nanotherapeutic point of view. In this chapter these methods will be described in greater detail. Due to the prevalence of electrospinning in NF fabrication, special attention will be paid to this production method.

### 2.1. Temple-Based Synthesis

Template-based synthesis uses the primary shape of the porous template material to confine the second material or the so-called precursor [16]. After filling the gaps of the template by the precursor, the template material can be either preserved or removed to give rise to hollow fibers. Widely used biocompatible materials for template synthesis of functional NF biomaterials are cellulose derivatives which provide structures with high-water retention capacity. Its hydroxy moieties can be easily functionalized for various therapeutic purposes [16].

### 2.2. Phase Separation

Phase separation technique is based on extracting solvent from a polymer gel [17]. Although simple and cheap method for the preparation of porous, continuous NF networks, it depends on the polymer type, gelation concentration, time and solvent. Moreover, it is time consuming, difficult to scale up and is not applicable to all polymers.

### 2.3. Self-Assembly

Self-assembly of amphiphilic polymers into 3D NF scaffolds is based on close mimicking of biological processes and functions of fibrous structures, and can be used for therapeutic purposes [18,19]. For example, some polypeptides can assemble via non-covalent and covalent interactions into fiber-like cylindric micelles and higher order structures such as ribbons and sheets. Judicious choice of amino acid sequences and their functionalization guide mimicking of physiological roles and enable control of charge, and physico-chemical and mechanical properties [20]. Similarly, under specific conditions amphiphiles of specific hydrophilic/hydrophobic segment compositions can assemble into high-aspect ratios such as NFs. Copolymers of polypeptides and hydrophobic tails can form NFs with sufficiently large hydrophobic cores to achieve high loading of hydrophobic drugs which can be superior to other type of nanostructures [21]. Other than hydrophobic interactions which can govern NF self-assembly, directional interactions such as hydrogen bonds, metal coordination or π-π aromatic stacking or electrostatic interactions between oppositely charged polymers can be exploited [22]. Other types of amphiphiles can self-assemble into NFs, including carbohydrate-based polymers [23,24].

Recently, stimulus-induced in situ self-assembly of peptides resulted in improved accumulation and prolonged retention times of NFs [25]. This assembly/aggregation induced retention (AIR) effect provides the basis for biological activity of self-assembling NFs and is increasingly exploited in theranostic therapies [26]. The mechanisms by which in situ NFs exert their biological activity are diverse, including assistance in drug delivery, blockade of tumor vascular system, cancer cell immobilization and interference of intracellular communication. In situ self-assembly extends the administration of NFs from the usual local delivery and activity towards intravenous delivery route for targeted stimuli-induced activity or therapeutic agent delivery. For example, the addition of acid-responsive poly(L-Lys) to doxorubicin-polypeptide conjugates enabled morphological transformation from spheric nanoparticles (NP) to NFs [27] (Figure 2).

While NPs remain stable under physiologic pH (7.4) and can extravasate into tumor tissue, the mildly acidic tumor microenvironment causes shedding of poly(L-Lys) and gradual NF formation with a prolonged tumor effect. Other physiological ques can induce in situ formation of NFs. Drug-carrying nanomicelles formed by multifunctional chimeric peptide, C16-K(TPE)-GGGH-GFLGK-PEG8, undergo a transformation from micelles to NFs after the cleavage by overexpressed pericellular cathepsin B [28]. Residual peptide labelled with cell-membrane targeting hydrophobic alkyl chain reassembles into NFs on the surface of the cancer cell membrane. The formation of “cell-in-shell” structure resulted in drug efflux reduction, overcoming of multidrug resistance and produced a synergistic therapeutic effect in a drug-resistant cell line. Overexpression of enzymes in tumor extracellular matrix (ECM) can also trigger local NF assembly by incorporating of enzyme-sensitive motifs [29,30]. Recently, complex ECM-inspired self-assembling NF probes for precise imaging of tumors by targeting fibroblast activation protein-α overexpressed on the surface of cancer-associated fibroblasts have been designed [31]. In a recent review paper, Liu et al. [32] have provided a detailed insight into in situ self-assembling nanofibers.

Another example involves the truncated amphiphilic peptide sequence derived from penetratin. It forms NF polyplexes with DNA, where DNA presents a template for β-sheet peptide agglomeration and NF growth [33]. The exposed hydrophobic amino acid residues interact with the cell membrane and are directly implicated in cell uptake.

### 2.4. Three-Dimensional Printing

Additive manufacturing or three-dimensional (3D) printing is a fast-growing technology used to produce layer-upon-layer structures with the support of computer-aided design (CAD). It can be performed in a variety of techniques in accordance with the used material. For example, techniques such as 3D-electroprinting, inkjet or extrusion printing can be used to prepare drug delivery platforms, prosthetics and tissue scaffolds. Since it is a relatively slow process, its main application is deemed to be in the area of personalized regenerative medicine, where it can meet the needs of individual patients by printing complex matrices [34], or tailoring DDS for individualized drug dosages and release kinetics. Cyto-compatible 3D printing methods such as extrusion, (stereo)lithographic, inkjet or laser printing can be used to fabricate hierarchical structures loaded with cells by direct printing of cell-containing polymer solutions (bioinks) and has a great potential in regenerative medicine [35]. The materials applicable to 3D printing need to fulfill specific viscoelastic properties in order to be printable and to maintain shape and mechanical fidelity. Moreover, they have to be accompanied by biocompatible methods for layer crosslinking [35,36]. Polymers such as polysaccharides, gelatin and silk are widely used as bioinks for creating matrices for cells or bioactive molecules.

In the context of nanofiber-based therapeutics, additive manufacturing is often combined with NF electrospinning to produce hybrid scaffolds with micro and nanomaterial features, and is especially investigated in tissue engineering [37]. The most often used approaches are layering of NFs and printed matrix and 3D printing of composite inks containing NFs.

For example, hybrid scaffold was produced by electrospinning of gelatin layer containing drug (hydroxyapatite, osteocalcin and type I collagen) on printed PLA scaffold [38]. Micro-sized grid exploits mechanical characteristics of PLA to provide structural support and gelatin, in turn, mimics the ECM of cartilage. The optimization of both processes is needed to insure required mechanical and structural properties. In electrospinning layer, polymer and drug concentrations were optimized. A 3D fused deposition modelling based on the extrusion of thermoplastic PLA through a temperature-controlled print-head required the optimization of scaffold morphology and pore size. In such composite scaffold, the ability of gelatin NF to form hydrogel under physiological conditions improved cell adhesion and provided mineralization. Similarly, biodegradable polycaprolactone (PCL) stent was printed on rotating mandrel and coated by dypiridamole-loaded PLA NFs [39]. As observed previously, deposition of NFs increased mechanical properties of the printed stent. In most of these examples, printed scaffolds of adequate mechanical properties provide structural support while NFs act as the therapeutic carriers. Whether and how do the printed components in these examples influence drug release is not known since drug release was only examined for NF components.

Temporalized release of biological queues loaded within the NFs can be programmed by different NF morphologies obtained by distinct electrospinning methods [40]. To achieve spaciotemporal release of two drugs from hierarchical composite, a multinozzle switching electrospinning subsystem was added to the 3D printing platform. Additionally, introducing a gradient of porosity along the thickness dimension into the printed gelatin/sodium alginate strut contributed to the control of desferoxamine (DFO) release from the intercalated layers of polyvinyl alcohol (PVA)-PCL core-shell NFs. Outer layers of this composite scaffold were composed of gentamycin sulphate (GS)-loaded PVA NFs. While GS underwent burst release from the outer layers within the first 4 days, DFO exhibited a sustained but stepwise release during 25 days corresponding to the gradient stent structure.

Biosynthetic approach based on genetic programmability of microbes can be used for the bottom-up design and sustainable production of therapeutic living materials. Inspired by fibrin nanofibers, genetically modified *E. coli* was used to produce proteinatious curli nanofiber biofilm by supramolecular crosslinking between α (knob) or β (hole) domains [41]. The obtained sheer-thinning hydrofilm composed of aligned nanofibers possesses good printability, shape fidelity and structural integrity. Therapeutic living architectures produced by hydrogels containing *E. coli* programmed to produce anticancer drug azurin demonstrated promising therapeutic applicability of programmable microbial inks.

### 2.5. Electrospinning

Among the different known methods for nanofiber preparation, electrospinning is the most utilized because of its simplicity. It has attracted significant research and business interest because of its flexibility and unique ability to deliver dry novel nanomaterials with controllable morphology and drug loading in a single step. Due to its overwhelming representation in scientific publications, we will look into it in greater detail.

A basic electrospinning setup (see Figure 1), comprises of three fundamental segments: a needle fitted with a metal nozzle, a collector on which a fiber is deposited and a high voltage supply which creates an electric field between the needle and the collector.

Polymer solution or melt is placed into a needle, and is extruded at a controlled flow rate to deliver a steady stream rate. To form a nanofiber, a high voltage (10–100 kV) is applied to the tip of the nozzle to produce electrically charged jet of polymer fluid. At a low voltage a spherical droplet is formed at the tip of the nozzle under the influence of surface tension forces. At the point at which the electric field arrives at a specific limit, the hemispherical charged surface of the fluid droplet expands and makes a tapered shape known as a Taylor cone. By further increasing the voltage, the basic value at which the applied external electrical stress exceeds the capillary stress of the polymer droplet (defined by its surface tension and meniscus radius) is reached, and the charged jet of polymer solution/melt spurts out of the Taylor cone towards the collector. The jet is consistently protracted under the influence of an electric field, creating a linear jet with a large surface area per volume and with diameters <1 μm which permits quick solvent evaporation or hardening of the polymer melt. This, in turn, leads to the formation of dry nanofibers and their deposition on the collector. At some point from the tip, the linear jet starts to be subjected to various instabilities, such as spiraling, coiling or bending, which influence fiber elongation and fluctuations in diameter [42]. Apart from these irregularities, bifurcation of the jet was also observed in some cases.

In order to minimize the irregularities and ensure homogeneity and reproducibility of nanofiber properties, modifications to the basic instrumental setup were introduced. The novel manufacturing setups and optimization of experimental parameters enabled the fine-tuning of fiber features and design of higher-order structures such as mashes, composites and 3D geometries, and have broadened the versatility and medicinal applicability of nanofibers.

To design the nanofibers for delivery of synthetic drugs and biopharmaceuticals, the nature of the therapeutic ingredient and the application, administration route and target site of the nano-formulation should be considered to ensure controlled and efficient therapeutic effects. The critical parameters that guide high loading and sustained release of therapeutic agents are closely related to nanofiber attributes and can be broadly divided into solution, processing and ambient parameters.

#### 2.5.1. Solution Parameters

The composition and physicochemical properties of carrier polymer have an important influence on the performance of the final nanofiber formulation since they are the major solid component of the fiber and define its safety, mechanical properties and matrix characteristics crucial for therapeutic efficacy. The influence of various polymer properties on the electrospinning process and fiber characteristics have been extensively evaluated.

The influence of polymer topology, concentration, molecular weight (Mw), molecular weight distribution (MwD) and other rheological parameters in the bulk and at the interface of polymer solutions on the electrospun fiber morphology was observed [43]. For solutions of linear polymers, the formation of continuous uniform fibers without beads-on-string and droplets was observed for concentrations at which sufficient chain overlap occurs [44]. For branched polymers of equivalent Mw, branch points tend to hinder chain overlap and entanglement, and bead-free nanofibers are spun at higher concentrations [45]. Additionally, the extensional viscosity of high Mw (>500 kg/mol) polymers can also contribute to the spinnability of polymer solutions [46]. In general, higher concentration and viscosity tend to produce nanofibers with higher diameter.

Although nanofibers have been prepared from natural polymer solutions, some characteristics such as limited solubility in organic solvents or ionogenic character of biomolecules such as polysaccharides lead to spinning inability due to high charge repulsion. Additionally, a three-dimensional network of hydrogen bonds results in high viscosity solutions even at low concentrations and prevents chain entanglement, adding to their limited applicability. Nonetheless, natural polymers such as silk fibroin, alginate and chitosan remain widely investigated due to high biocompatibility and biodegradability. By blending them with non-ionic, flexible hydrophilic synthetic polymers such as poly(ethylene oxide) (PEO) or poly(vinyl alcohol) (PVA) at appropriate ratios their spinnability and mechanical properties can be enhanced [43,47]. Electrospinning of biomolecules is also possible by judicious choice of organic solvents, but this can be related to the need for additional solvent removal when volatile solvents such as dichloromethane are involved [48,49].

The solute-depending factors influencing the electrospinning outcomes interplay heavily with other solution properties. Solvent characteristics such as the ability to dissolve the polymer are combined with volatility, polarity or dielectric constant and can determine the spinnability of the solution as well as morphology and orientation of the fibers [50,51,52,53]. One of the parameters that is indispensable for successful electrospinning is the solution conductivity. Dielectric polymer solutions cannot be spun because of the lack of charge separation needed for Taylor cone development. The extensive research on the influence of solution conductivity determined that spinning of solutions with higher conductivity results in nanofibers without beads and with lower diameter. The addition of salts to the polymer solution serves as a source of ions necessary for the formation of the Taylors cone and increases conductivity simultaneously. However, excessive conductivity impairs cone formation due to a reduction in the tangential electric field [54]. The choice of salt with a given physicochemical properties (size and structure) and concentration in a combination with the physicochemical properties of a solvent and proper polymer can influence not only the properties of the nanofiber (diameter and morphology), but also the spinning performance. In a free surface spinning that uses a rotating roller electrode, complex interactions between polymer, solvent and salt molecules influence not only the number of created cons and jets on the spinning roller, but also the overall performance and the quality of spun fibers and webs [55]. Conductivity od polymer solutions in coaxial spinning can influence the diameter ratio of core to shell and diameter distribution as observed for composite core-shell fibers composed of cellulose acetate-PCL and chitosan. While higher conductivity of the core polymer (PCL) forced fast core elongation producing thinner fibers with lower core-to-shell ratio, lower conductivity resulted in partial permeation of layers [56].

#### 2.5.2. Processing Parameters

An array of processing parameters should be adjusted in order to achieve NFs of desired physicochemical and therapeutic properties, and parameter optimization presents a routine part of the electrospinning optimization process [57,58]. Applied voltage depends on solution conductivity, surface tension and viscosity, solvent volatility and the right choice of solvents and polymer solution concentration is of paramount importance [59]. Like in the case of flow rate, an increase of voltage above a critical value will lead to changes in NF morphology, and deformations such as bead formation, increase in NF diameter and pore size can be observed as well as wide distribution of diameters due to unstable spinning process [60]. Changes in the voltage supply can direct the shape of NFs and the final NF mat products. While the most electrospinning processes are performed under direct-current voltage, pulsed voltage (PV) of given strength, duration and frequency can be used. PV can lead to the formation of thinner NFs of adjustable length. It was observed that, at least in some cases, increased process stabilization and better control over NF mat morphology could be provided by applying PV [61]. Other parameters such as nozzle tip-to-collector distance should be optimized to enable solvent evaporation and NF formation since residual solvent can lead to fusion of fibers and changed structure of resulting mat. In general, longer distance results in finer fibers.

The collector is another key element of the experimental setup that can influence process productivity and the properties of the final product. The collector should efficiently ground the built-up charge of depositing fibers, and the choice of grounding electrode has a decisive influence on the number of deposited NFs. Fiber deposition is encouraged by the reduction of charge density of the jet and dissipation of residual charge, which can be accomplished by introduction of focusing and stirring electrodes. Additionally, using collectors with uneven topology can result in uniform fiber deposition, which could be significant in mass production [62]. They can also be used to produce patterned mashes or grid membranes. Similarly, moving collectors like the one based of spirograph pattern can result in deposition of NFs with improved mat thickness, mechanical properties and porosity, as well as improved NF orientation [63]. In cases when fibers cannot dry sufficiently, coagulation baths can be used as collectors. While static collectors usually result in randomly oriented mats, rotating collectors such as drums or disks can be used for the deposition of aligned fibers [64]. Fiber orientation can also be achieved by multi-segmented collectors [65]. Although the before-mentioned collector configurations can produce 2D mats and membranes, electrospinning of 3D biomimetic curvilinear structures often needed in regenerative medicine is still challenging. New grounded collector based on hydrogel scaffold diversified the applicability of electrospinning towards new complex macroscopic 3D structures [66]. In this case, the reversible sol-gel transition of gelatin was used to partly remove the hydrogel scaffold which resulted in hollow nanofiber structure. Biocompatible nature of hydrogel also permitted the incorporation of biomolecules and living cells, creating composite NF biomimetic structures for slow release of biomolecules and scaffolds for regenerative tissue application.

Like other nanodelivery systems, the NF formulations intended for drug delivery face the need to preserve the functionality of biopharmaceuticals. In that regard, electrospinning has some crucial advantages when compared with freeze-drying or techniques based on spray drying since moderate drying temperatures are involved. However, single-nozzle electrospinning configurations widely used in laboratory setups are slow (0.5–1 mL/h) and do not meet the requirements of commercial scale-up NF production. The solution to the problem was the introduction of multiple needles. However, this approach suffers from the electric field distortion due to the presence of multiple jets [67]. Another approach is needle-less free surface electrospinning, which can suffer from changes in solution composition and viscosity. A recent attempt to account for the drawbacks of the previously mentioned methods was the introduction of a fast-rotating atomizer multi-hole spinneret which increased the maximum feeding rate (40 mL/h, 6 g/h) when compared to syringe and needle-less electrospinning [68,69]. Omer et al. explored the latest advances in design of electrospinning equipment for large-scale NF production [5].

#### 2.5.3. Ambient Parameters

The interplay of environmental temperature and relative humidity can produce nanofibers of reproducible characteristics. In general, an increase in temperature reduces surface tension and solution viscosity and allows for the formation of uniform fibers [70]. Various studies have observed that the impact of relative humidity (RH) on fiber diameter depends on the nature of the used solvent and their interaction with the environment. Slow evaporation of water solutions under the regime of high RH leads to increased fiber elongation and results in finer fibers. When organic solvents are used for water-insoluble polymers, water diffusion which can occur under high RH conditions can cause polymer precipitation and reduce jet charge leading to thicker fibers. Therefore, the RH can influence fiber morphology depending on the water-miscibility of organic solvent. For example, high RH causes increased formation of small circular pores when water-immiscible solvent was used to spin poly(caprolactone) (PCL) fibers. In contrast, with increased content of tetrahydrofurane (THF) crater-like surface features were observed. In both cases, an increase in environment temperature delayed changes in morphology observed for higher RHs [71]. However, the appearance and the shape of morphological features in the moisture-sensitive polymer solutions composed of hydrophobic polymers in a water-miscible or polar solvents, strongly depends on the interplay of solvent and solute properties and their interaction with the external conditions of RH and temperature [72]. 

The temperature of working fluids can also influence the spinning process. The solubility of some semicrystalline polymers as well as the additives such as low-solubility drugs used during the electrospinning process is elevated when the solution temperature is increased. Additionally, the elevated temperature decreases solution viscosity and surface tension and facilitates fiber formation, which could be relevant in spinning polymers with very high Mw. As observed in the case of polyvinylpyrrolidone (PVP), increase of solution temperature influenced fiber diameter and surface morphology. Reduced resistance to fiber formation with temperature increase led to the formation of thinner fibers, while faster evaporation of a less volatile solvent (*N*,*N*-dimethylacetamide) resulted in fibers with smoother surface [73]. However, excessive solvent evaporation at too high temperatures could prevent fiber formation, and depending on the instrumentation design accumulation of highly volatile solvent vapors within the injection syringe could intermittently interrupt the spinning process. However, in the case of highly volatile solvent (ethanol) smooth fibers with higher radius were obtained [73]. A recent study on the electrospinning of chitosan (Cs) demonstrated that the optimization of both solution and the environmental temperature might produce uniform nanofibers [74].

### 2.6. Blow-Spinning

Solution-blow-spinning (SBS) or airbrushing is another low-cost method of NF preparation. Unlike much-used electrospinning techniques, airbrushing is fast and requires a simple experimental setup that could be used for the on-site NF fabrication and deposition. Therefore, it is a promising technique that could be used for direct patient application as required in would healing or tissue engineering. Additionally, the method is well suited for scaling to industrial production requirements with easy adjustment for different processing configurations [75,76,77]. Despite the growing interest in this production method, there is still a limited number of publications related to drug delivering SBS NFs.

This method is based on extrusion of polymer dissolved in a volatile solvent by pressurized gas through a spinneret. The outer nozzle that surrounds the polymer-expulsing one uses the same source of pressurized air for solvent removal. Shear forces at the interface of gas and polymer solution result in solution stretching and NF formation. Instead of concentric nozzles, commercially available airbrush can be used to form polymer stream. Finally, NFs are formed on collector. Device design and nozzle setups allow the production of single-compartment or core-shell NFs which can be of random or parallel orientation. Additive deposition allows formation of customized 3D scaffolds. Unlike the much-used electrospinning, the apparatus setup is simpler and does not require the use of electric field. Additionally, it is much faster than electrospinning.

Experimental parameters such as polymer molecular weight, solution concentration and viscosity influence NF formation [78]. Additional experimental variables such as flow rate as well as air pressure can influence structure and morphology of NFs [79,80]. Formation of NFs in SBS is governed by the same theoretical principles as in electrospinning, emphasizing the need of the minimum polymer Mw and concentration [81]. Furthermore, similar morphological transitions including beads-on-string were observed before the formation of uniform NFs. The optimization of polymer feed rates and gas pressure are necessary for the production of the stable polymer jet and control of NF diameter and well-defined morphology. Although SBS experiments are usually performed at room temperature, it can be manipulated to improve solution viscosity or polymer solubility. Similarly, surrounding temperature, humidity and working distance can present parameters in SBS. Unlike electrospinning, 3D scaffold produced with similar polymers by SDS have higher porosity and pore size which might be beneficial for cell infiltration. However, tighter diameter distribution was observed in SDS. Distinct differences in mechanical characteristics of produced NFs were also observed between the two methods [81,82].

Comparison of production techniques was demonstrated on example of diclofenac inclusion and release from poly(3-hydroxybutyrate-co-3-hydroxyvalerate (PHVB) NFs prepared by electrospinning and SBS of hexafluoropropanol solutions [83]. Although drug loading resulted in increased NF diameter, it was smaller in SBS NFs. While shear forces dominate fiber stretching in SBS, fiber formation in electrospinning is dominated by surface tension overcoming. Moreover, the speed of fiber formation is likely to have influence on polymer molecule orientation within the fiber and influences the optical properties of fibers. Drug release rate was similar in both cases.

Influence of model drug physico-chemical properties on morphology of SBS corn zein protein NFs demonstrates interactions with polymer domains and more ordered structures [84]. In this case drug inclusion did not influence NF diameter or drug release profiles. However, the loss of ordered fibril structure was observed and the lack of mass loss indicates possible diffusion-governed drug release. In another model study, addition of hydrophilic porogen to core-shell polyvinylpyrrolidone (PVP)-PCL NFs caused the burst release of hydrophilic drug from the hydrophilic core [14]. At the same time, leaching of hydrophilic domain from hydrophobic shell created pores and increased wettability and surface roughness, resulting in increased cell proliferation.

SBS can also be combined with other techniques. The combination of blow and centrifugal spinning resulted in a novel pressure-driven method called pressurized gyration (PG) with the potential of a scale-up NF production [85,86]. In PG, the polymer solution is extruded through the holes in a drum under the influence of high rotation speed and applied pressure followed by fast solvent evaporation and collection of NFs on the surrounding plate. Compared with a benchmark gold standard electrospinning process, PG produces NFs of distinctly different morphologies and more specific drug release behaviors, with higher production rate and absence of nozzle clogging that can plague the practicality of electrospinning. Recently, the method was used to incorporate hydrophobic drug (progesterone) into hydrophilic NF [87]. Parameters such as solvent and polymer composition as well as drug loading were optimized to achieve necessary viscosity and surface tension for spinning and the required NF morphology. Obtained mucoadhesive NFs demonstrated a suitable drug releasing profile for potential use as vaginal inserts in prevention of pre-term birth. Moreover, the method can be adjusted to produce core-shell NFs, and parameters such as working pressure and rotating speed had influence on NF size and surface morphology [88].

### 2.7. Melt-Blowing

Melt-blowing is a conventional solvent-free technique for NF fabrication and can be applied at industrial scale. It is based on the extrusion of a polymer melt through a small-size nozzle surrounded by a high-speed blowing gas. This technique was successfully used for the production of blend NFs composed of hydrophilic vinylpyrrolidone-vinyl acetate copolymer (PVPVA64) and PEG 3000 plasticizer for oral delivery of the poorly water-soluble drug carvedilol [89]. However, the possible decomposition of polymer or therapeutic agents present one of the limitations of this technique.

### 2.8. Other Methods

Agricultural biomass and renewable waste present abundant sources for production of biodegradable and biocompatible cellulose NFs (CNF). Various processing methods have been used to obtain CNFs. Compared to initial methods based on chemical hydrolysis of biomaterial which demand prolonged processing steps, more eco-friendly high-yielding enzymatic hydrolysis was introduced [90]. Different enzymes can be used to control the site of hydrolysis and can be combined with other processing techniques. Functional carboxylic groups of CNF can be functionalized by grafting in order to optimize drug loading [91]. Apart from plant sources, microorganisms can also be used to produce CNF. Bottom-up production of cellulose by bacteria yields BC with higher degree of crystallinity and purity than that isolated from plants, and has been scaled to industrial production [92]. 

## 3. Delivery of Therapeutics

Nanofibers possess advantageous features such as great surface area, high loading capacity, ease of functionalization and the ability to simultaneously incorporate diverse therapeutic agents which make them attractive drug delivery systems. Additionally, the variety of instrumental setups and controllable experimental conditions enable design of tunable morphologies for optimized drug loading and release. Adjustment of mechanical properties of NFs enables drug delivery through various administration routes, and oral, topical, transdermal and local drug deliveries were explored. So far, different therapeutic agents have been delivered for the treatment of a wide range of diseases and conditions, and include antimicrobial and anticancer small synthetic molecules, nucleic acids (DNA and RNA), peptides and proteins, as well as whole cells such as bacteria or stem cells used for tissue regeneration. 

Active pharmaceutical ingredients (API) can be incorporated within the nanofiber network during manufacturing process, or post-processing by physical or chemical bonding to the nanofiber surface.

The simplest method of drug incorporation within the NF is by processing drug-polymer solution blend, as often encountered in spinning and 3D printing methods discussed in Section 2. A significant number of anti-inflammatory, antibiotic or anticancer medicines currently in clinical practice or in the pipe-line are of low water-solubility, and incorporation within nanofibers presents one of the methods to increase drug solubility. Differences in physico-chemical properties between polymers and therapeutic load, namely solubility or susceptibility to degradation and denaturation present challenges when preparing NFs loaded with biological therapeutics such as enzymes or therapeutic proteins [93]. The solution presents itself in co-axial electrospinning which separates inner and outer solutions. However, applied voltage, shearing forces present at the interface, and rapid dehydration can compromise load stability and functionality, and the presence of multiple phases can influence NF consistency [94]. Alternatively, colloid electrospinning of oil-in-water emulsions is a more appropriate choice of method that can be used for small drugs, proteins and enzymes. NF scaffolds loaded with functional protein can also be prepared by spray nebulization [94].

The high porosity and surface area of NFs can be used to adsorb therapeutic molecules on the surface through a variety of weak non-covalent interactions such as electrostatic, hydrophobic or hydrogen bonding. Such rather weak interactions can cause relatively fast drug release from the surface of NFs. To improve drug release profiles, various strategies were developed. Recently, composite cellulose acetate(CA)/poly(ε-caprolactone diol)/poly(tetramethylene ether)glycol-polyurethane NFs with incorporated multi-walled carbon nanotubes (MWCNT) were used for Dox adsorption and were applied to prostate cancer cell treatment [95]. The addition of MWCNT to the NFs led to significantly improved sustained Dox release as well as cancer cell death when compared to NFs without nanotubes.

Functional groups of NFs present the base of another major strategy to introduce therapeutic agents on nanofiber surface. Functional chemical moieties can be activated to create stimuli-sensitive linkages with therapeutic molecules or other drug carriers. This principle was used for loading chitosan NFs with a hydrophobic drug. Since chitosan NF cannot be loaded with hydrophobic drugs, the amino groups of electrospun chitosan NFs were conjugated with β-cyclodextrin (β-CD) via amide bond without the loss of NF structure or physicochemical properties [96]. The ability of CD to form inclusion complexes with lipophilic drugs such as dexamethasone resulted in improved drug loading and superior sustained release when compared with unmodified NFs (12 vs. 3 days for complete drug release). Similarly, NFs surface can be modified to adsorb proteins and enzymes with retention of a secondary structure and functionality, potentially extending their application to adsorption of therapeutic biomedicines [97]. Recently, avidin moieties conjugated on the surface of hydroxyapatite-coated gelatine NFs were used to bind biotinylated growth factors. The strong non-covalent avidin-biotin interactions significantly improved the sustained release of loaded biomedicines and resulted in enhanced bone regeneration [98]. Another surface functionalization method that enables the preservation of functional properties of immobilized biomolecules is the layer-by-layer (LbL) self-assembly technique. Alternating deposition of interacting layers on top of NFs creates a multi-layered reservoir for various types of active pharmaceutical ingredients. Multiple layers composed of positively charged poly-L-lysine (PLL) and negatively charged alginate on the surface of PCL NFs served as a medium for immobilization of Inactivated Hemagglutinating Virus of Japan Envelope (HVJ-E) [99]. Additionally, to improve stability under physiological conditions, layers were crosslinked via covalent bonds. HVJ-E was efficiently desorbed under physiological conditions of ionic strength and temperature and was able to suppress cancer cell proliferation. The advantages and drawbacks of different post-modification methods currently used to functionalize NF surface were systematized in several recent review papers [100,101,102] and will not be pursued in greater detail here.

In the case self-assembled NFs, drug loading is achieved in similar fashion. Therapeutic load can be a constitutive component of multidomain polymer. Noncovalent interactions between polymer domains and therapeutics are also widely exploited. Electrostatic interactions between cationic segments and polynucleotides, and interactions such as Van der Waals, hydrophobic and hydrogen bonds can be exploited to load small molecular drugs. Detailed examples are presented in Section 2 and Section 4.

Bioavailability and biodistribution, and therefore the therapeutic and side effects, depend on physicochemical properties of drugs, such as molecular weight, solubility, polarity, hygroscopicity and stability. Depending on the treated condition and the administration route, the drug delivery system should be able to release controlled amounts of therapeutic agent for a defined period of time.

A number of parameters can influence drug release. Apart from the chosen polymer parameters, processing parameters can also influence drug release and should be optimized on a case-to-case base. Undesirable drug release mechanisms related to burst release, uncontrolled release kinetics or incomplete drug release can compromise therapeutic effects and be a source of unwanted off-target and side-effects. While underdosage can reduce desired therapeutic effects, overdosing can lead to side-effects. Moreover, excessive delay in drug release can cause local toxicity and drug resistance. Another aspect of drug release from the nanofibers is the influence of fiber degradation on the drug release profile. Therefore, a long-term release studies are necessary to evaluate the biological effects of drug releasing profiles.

Physical processes that govern drug release related to drug surface desorption, diffusion, and NF and composite matrix degradation are common to various fabrication methods. Some of related parameters that influence drug release have already been described in Section 2. Due to the prevalence of scientific reports related to electrospinning, the majority of existing information on how to manipulate and exploit these mechanisms was obtained for this manufacturing method. However, similarities could be observed in comparison studies involving electro and solution blow spinning (Section 2.6).

Polymer blends are often used to modulate physicochemical properties and biocompatibility of nanofibers in order to ensure their performance under highly complex physiological conditions. Blending natural and synthetic polymers with different flexibilities and polarities can provide scaffolds with improved mechanical properties, bio-affinity, drug diffusion and release profiles. The hydrophilicity, matrix degradation rate and drug release from poly(D,L-lactide-*co*-glycolide) NFs was regulated by spinning with different amounts of poly(ethylene glycol) (PEG), poly(ethylene glycol)–*b*-poly(d,l-lactide) (PELA), polyglycolide (PGA), poly(dioxanone) (PDO) and poly(trimethylene carbonate) (PTMC) [103]. During the life-time of the NF membranes several drug-releasing phases were observed. Since polymers and the model drug were well soluble in the solvent, regular smooth-surface fibers exhibited homogeneous drug distribution throughout the fibers. Under physiological conditions, in the initial stage I fibers swell in contact with PBS and initial burst drug release was observed, predominantly from the fiber surface. With the prolonged contact, further swelling of fibers can cause increase in fiber diameter (Figure 3F), fiber fusion and decrease in inter-fiber spaces which forced prolonged drug diffusion towards the exposed surfaces and slowed down drug release in stage II. In stage III, degradation of polymers into oligomers and the subsequent weight loss created large pores in dense films obtained by swelling and liberated large amounts of trapped drug. The higher the content of hydrophilic polymer (PELA), the shorter absorption time of water and the greater initial drug release from the nanofiber membrane surface (Figure 3D,E). In this case, less swelling in stage II caused more gradual transition between drug release in stages II and III resulting in more sustained drug release (Figure 3A–D). This basic three-stage drug releasing process was observed for other polymers blended with PLGA, except with PDO where linear, a relatively fast drug release was observed. The release profiles, however, differ in accordance with the complex interplay of polymer water solubility, polarity, the ability to form chain networks, and degradability. For example, while complete release from PDO occurs before polymer degradation, release form PLGA/PGA is dictated by the earlier onset of stage III for the PGA component, while most of the drug from PLGA is released in stage III. The practical consequence of such differential drug release could be exploited to formulate layered nanofibrous structures with optimized prolonged drug release.

Controlled and sustained release of therapeutic load from nanofibers is crucial for the maintenance of therapeutic dose and minimization of potential side-effects caused by high burst-release drug concentration. Drug release can also be controlled by selection of polymers with appropriate crystallinity, wetting and glass transition temperature (Tg). For hydrophobic polymers where wetting and diffusion of water are limited, drug release tends to be slower than in the case of hydrophilic polymers. Crystallinity or the proportion of ordered, crystalline segments of polymers compared to amorphous, randomly ordered regions can influence water penetrability into the fiber. Interactions between different polymers or between polymer and drug load can influence Tg, the temperature at which polymer transits from rigid to a more flexible state due to increased chain mobility. Such a decrease in Tg can result in increased drug release and can serve as a base for design of stimuli-sensitive drug release under physiological conditions. An Eudragit^®^ RS 100, a copolymer of poly(ethyl acrylate, methyl-methacrylate and chlorotrimethyl-ammonioethyl methacrylate) and poly(methylmethacrylate) (PMMA) blend was tuned to release loaded molecules as a response to thermal stimuli when nanofiber membranes were exposed to physiological temperature [104,105]. Glass transition temperature of 34.8–36.5 °C in water environment (“wet” Tg) enabled the release of antimicrobial Octenidine that was 8.5-times higher in the “on” state than under the “off” conditions (25 °C), and could be repeated over five cycles [104]. In the infected areas where the wound temperature is higher than the temperature of surrounding healthy skin, such tunable nanofiber wound dressings are ideal depots for on-demand release of therapeutic agents for wound healing.

The ability of PNIPAAM NFs to form hydrogels by incorporating high amounts of water in their 3D network makes them attractive soft tissue-mimicking systems for controlled drug delivery. Adjustment of lower critical solution temperature can be used to trigger drug release from thermoresponsive polymers such as poly(N-isopropylacrylamide) (PNIPAAM). Bellow the LCST (32 °C), intermolecular hydrogen bonds between the polymer and aqueous solvent contribute to high polymer hydration and swelling. Above that temperature, the formation of intramolecular hydrogen bridges leads to polymer precipitation and drug release, which can be additionally fine-tuned by UV-induced shell crosslinking [106]. The addition of more hydrophobic copolymerizing component such as *N*-isopropylmethacrylamide (NIPMAAM) to NIPAAM can tune the LCST closer to 37 °C and enable programmed drug release under physiological conditions [107]. Electrospinning of water-like P(NIPAAM-NIPMAAM) precursor solutions has demonstrated to be challenging since they are composed of small molecules and rarely meet the critical conditions necessary for successful electrospinning process. Encasing hydrogel P(NIPAAM-NIPMAAM) shell within the polymer core by co-axial electrospinning with concomitant UV hydrogel crosslinking was used to prepare stimuli-sensitive drug releasing NF system. Special care should be taken in controlling environmental conditions during electrospinning of these NFs, since too high temperature could induce undesired hydrogel conformation transitions and too low humidity the loss of aqueous solvent. Poly-L-lactide-*co*-caprolactone (PLCL) core serves as a porous membrane that can regulate drug release from NFs. Additional control of drug release by additional adjusting of Tg and the thickness of PLCL shell resulted in improved release sustainability during 41 days.

Controllable stimuli-responsive drug release is in close relationship with other parameters that guide drug release from nanofibers, and is a function of nanofiber morphology and physico-chemical properties of therapeutic agents [108]. Majority of thermosensitive polymers are often based on acrylamide derivatives which can be blended or combined with other polymers into a variety of morphologies. Some other thermosensitive NFs are based on polycaprolactate derivatives such as poly(di(ethyleneglycol) methyl ether methacrylate (PDEGMA)/poly(l-lactic acid-co-ɛ-caprolactone (P(LLA-CL) blend [109] or poly(N-vinylcaprolactam) (PNVCL) [110]. Unlike PNIPAAM, PNVCL does not produce toxic compounds during hydrolysis-mediated degradation.

Apart from exploiting thermosensitive capacity of some polymers, other external stimuli can be used to change physical or chemical properties of NF carriers as the “on/off” switches for controlled drug release. Changes in extracellular pH or redox state, light or presence of certain biomolecules can trigger changes such as polymer sol-gel transition, precipitation or dissolution. Polymers that respond to different stimuli can also be combined into blends or composite NFs as demonstrated in Table 1.

Similarity of nanofibers with the microporous structure of extracellular matrix (ECM) is often exploited as ideal support for cell migration and growth necessary for tissue regeneration. As such, they can be applied as carriers of growth factors with controllable release properties sensitive to external stimuli, such as the changes in ECM redox. Incorporation of GSH-sensitive polyethyleneglycol-polycaprolactone/6 arm polyethyleneglycol-polycaprolactone-sulfhydryl nanogel into a PCL shell of polyethyleneglycol (PEO) core loaded with bone morphogenetic protein 2 (BMP-2) led to formation/sealing of nanochannels within the shell due to the presence of sulfide moieties [115]. The response to the changes in physiological GSH concentrations led to improved bone healing in in vivo model. Lability of polyoxolate (POX) towards H_2_O_2_ was exploited to prepare composite polyvinyl alcohol (PVA) NFs for treatment of inflammatory diseases characterized by increased H_2_O_2_ production. Release mechanism and kinetics of drug release in this case can be controlled by the amount of POX. While at low POX ratios release is fast due to PVA solubility and is governed by diffusion, at higher ratios of hydrophobic POX release is more sustained and is dictated by H_2_O_2_-triggered NF degradation [122].

The addition of pH-responsive components to NFs can increase and prolong their stability, and improve sustained drug release. For example, collapsing of the pH-sensitive 4-vinylpyridine shell component provided protection to PVA core at pH > 6.5 [114]. At the same time, collapsed coating resulted in smaller NF radius, increased free volume and, therefore, improved drug release than observed at pH 4. Therapeutic effects can also be achieved by the incorporation of functionalities that hydrolyze under certain pH conditions. For example, liposomes were grafted on oriented polylactic acid (PLA) fibers by acid-sensitive Schiff base bonds which additionally led to the stabilization of fibers by crosslinking [9]. Partial fusion of aminated NFs and liposomes also improved mechanical properties of biomimetic NFs that are crucial to meet the demands of physiological activity of spine and the surrounding tissue. Such functionalized NFs were used for simultaneous delivery of different therapeutic species (Figure 4).

Oriented fibers posed as directional scaffold for the growth of axons and were at the same time loaded with the therapeutic neuron growth factor while the grafted cationic liposomes served as reservoirs of therapeutic pDNA. After dissociation under the acidic conditions that is typical for an inflammatory response after spinal cord injury (SCI), cationic surface of liposomes promoted their cellular uptake. In the case where NFs are used for bone or tissue reconstruction, the nanomaterial should degrade gradually to allow sufficiently long support for tissue regeneration. While fast release (10 days) of liposomes and presentation of therapeutic interleukin is necessary for regulation of SCI immune-environment, slower release of therapeutic nerve growth factor was accomplished for over 40 days. The formulation was successfully tested in animal model, resulting in reduced inflammatory response, increased nerve repair and recovery of motor function.

Polyesters are often used as enzyme-sensitive NF components, but their hydrolytic or enzymatic biodegradability tends to compromise mechanical properties of NFs. On the other hand, introduction of amide moieties can improve both mechanical and thermal characteristics of polymers and introduce new cleavable positions. Ester-amide copolymers (coPEA) are increasingly explored as components of NFs which exploit overexpression of ECM enzymes for targeted drug release [93]. Blend coPEA/PCL NF were loaded with aqueous droplets containing enzyme and model flurophore. The loaded enzyme served as an internal stimulus which can additionally contribute to the matrix degradation by external enzymatic stimuli and influence drug release kinetics. Other examples of enzyme-responsive drug release include thiolated hyaluronic acid (HA) sensitive to seminal hyaluronidase (HAse) for intravaginal release of a water-soluble nucleotide reverse-transcriptase inhibitor tenofovir (TVF) aimed at anti-HIV therapy [116]. The application of mucoadhesive HA NFs had no adverse effects on organs, epithelial cells of reproductive system or vaginal microbiome, and provided higher drug bioavailability in the presence of HAse than the control TVF-gel formulation in vitro.

Some polymers such as biocompatible polyvinylidene fluoride (PVDF) possess piezoelectric properties which can be used for controlled drug release by converting applied mechanical force into electric potential to release molecules adsorbed on the surface of piezoelectric materials. Intrinsically low piezoelectricity of PVDF was enhanced by downscaling the blend polymer polyvinylidene-trifluoroethylene (P(VDF-TrFE)) into nanofibers. The ability to adjust the sensitivity of NFs to the magnitude and frequency of the applied external force by controlling NF diameter resulted in controllable release of electrostatically adsorbed drugs [120].

Another approach to design of smart NFs is to prepare composite nanofibers by introduction of stimuli-sensitive nanoparticles within or on the surface of nanofibers. The ability of Au nanorods (AuNR) to absorb NIR light (650–900 nm) and to generate heat due to the surface plasmon effect was used to prepare theranostic PNIPAAM NFs [117]. Deep biocompatible tissue penetration (up to10 cm) was used to trigger reversable temperature-related phase transition in PNIPAAM and subsequent drug release. Additionally, PNIPAAM fibers were cross-linked to improve the aqueous stability of homopolymers. In vitro experiments on malignant glioma cells demonstrated that the NIR photothermal “on-off” could be applied for the localized sequential release of drug and is dependent on NIR laser pulse strength and duration. Although this approach could overcome the issues related to local chemotherapeutic toxicity and drug diffusion limitations, its applicability could be limited by the tissue penetrability of applicated irradiation. Irradiation-based drug release could exhibit more practicality in skin treatments or transdermal drug delivery. Mild visible light stimulus was used for on demand release of drug from a transdermal DDS by incorporating VIS-light responsive goethite nanoparticles loaded with drug (α-FeOOH) into the core of NFs [118].

Composite NFs can also be prepared for drug release stimulated by magnetic field. Actuation by external magnetic field holds a great promise in drug delivery and therapy since it enables controlled and precise manipulation. For example, magnetic iron oxide nanoparticles (IONP) are often used for preparation of stimuli-responsive NFs [119,121].

Implantable IONP-based NFs have the potential antitumor application by combining hyperthermic effect caused by application of external magnetic field with chemotherapy as described in the case of electrospun PLGA NFs functionalized by a shell-mimicking mussel adhesive dopamine. Catechol amino groups of dopamine have high affinity for metal oxides and the ability to coordinate borate-containing compounds such as bortezomib which can be released under reduced pH of tumor microenvironment. Simultaneous combination of dual therapy exhibited synergistic effect and induced apoptosis in breast cancer cells (4T1).

The formation of flexible nanofiber mats that can conform to skin can be integrated with electronic devices that serve as heat source for external stimulation of drug release. Elastic NF mash was prepared from poly(glycerol sebacate)-poly(caprolactone) (PGS-PCL) blend containing thermo-responsive PEGylated-chitosan nanoparticles (NP) loaded with antibiotics without the loss of mechanical properties [113]. Stretchable conductive patterns were printed on the surface of elastic mashes by radio frequency sputtering of various metals. The use of printed metal devices that are degradable at the alkaline physiological skin wound pH results in biocompatible and biodegradable DDS with great potential for topical application. Since the sol-gel transition temperature of chitosan NPs is above skin temperature, controlled temporal release of antibiotics was achieved by mild increase of temperature (38–40 °C). However, to avoid uncontrolled release of drugs caused by fluctuations in external temperature, NF mats should be thermically insulated. Further adjustment of polymer transition temperature might extend the applicability of such devices to internal implants.

## 4. Therapeutic Applications

### 4.1. Delivery of Small Chemotherapeutic Molecules

The ability of NFs to be tailored for controlled delivery of both hydrophilic and hydrophobic drugs is widely used within the universe of drug nanocarriers. The wide choice of polymer materials and production techniques is increasingly used to produce drug-loaded NFs that can be applied by various administration routs (Table 2).

Nanofibers composed of self-assembling amphipihilic peptides (PA) are attractive nanocarriers due to simple preparation method and biocompatibility. They can be used for delivery of wide range of therapeutic molecules via diverse delivery routes. Intravenously administered therapeutic nanostructures present attractive strategy to reduce plaque burden and fight cardiovascular diseases. Self-assembled peptide NFs labelled with apolipoprotein A1 targeting sequence were used to deliver Liver X Receptor agonist GW3965 to atherosclerotic lesions [123]. The particular shape and increased surface area of NFs when compared to spheric nanoassemblies facilitate their interactions with blood vessel walls which combined with the ability of targeting F4 moiety to bind to oxidized lipids of plaques leads to efficient plaque burden reduction but without adverse effects of the free drug. Self-assembling PA NFs can be applied for local drug delivery for tissue regeneration. A recent study reported functional recovery after spinal cord injury by Taxol delivered by PA NFs decorated by neural cell adhesion molecule motive [124].

Electrospinning is another method widely used to prepare drug-loaded NFs for therapeutic purposes and the overwhelming number of studies related to delivery of small chemotherapeutic drugs is related to the use of this production method. Different modalities of electrospinning process provide a wealth of NF morphologies that can be optimized to suit the encapsulation of single or multiple pharmacological agents and desired release profiles and mentioning them all is beyond the scope of this review. However, we will present a few of the most recent concepts. A new type of improved wound sutures was designed by electrospinning to address poor drug loading and inadequate release kinetics of existing sutures. Although nanofibers produced by electrospinning possess favorable drug loading and release kinetics, the mechanical properties needed for sutures are far from ideal. The aggressive post-production methods that are needed to improve mechanical properties could compromise the stability of therapeutic molecules and therapeutic outcomes. The solution to overcome these problems was found in assembling continuous electrospun core-yarns followed by deposition of electrospun drug-loaded sheet [129]. Mechanical strength of such yarns was provided to the PLA core by twisting the continuous fibers over the rotating bobbin and heat-stretching them at the optimum temperature. Wound core was subsequently uniformly coated by a blend solution of PLGA and anti-inflammatory drug aceclofenac or insulin, after which the yarns were double plied to form sutures. In these core-sheet yarns, the sheet maintains the properties of superior drug loading/release characteristic for electrospun NFs combined with the mechanical properties of pre-processed NF core. The initial burst release of therapeutic agents from these core-sheet yarns is attenuated and sustained release was observed for both therapeutic agents (10 days for aceclofenac and 4 days for insulin). Released insulin successfully promoted fibroblast migration in vitro, while the application of aceclofenac-loaded sutures in vivo attenuated inflammation. Unlike some commercially available degradable polymer sutures which can cause mild inflammatory response due to the surface lubricant, these yarns do not require the use of such lubricants. Recently, antibacterial suture yarns loaded with tetracycline hydrochloride were produced by dual-electrospinning method where chitosan/polyvinyl alcohol and PLA solutions were concomitantly injected from separate syringe [133]. Mechanical properties of woven NFs make them amenable to weaving into implantable drug-loaded nanotextiles. Polydioxanone (PDS) nanotextiles showed a sustained prolonged release of Ptx during 2–3 months, controllable by the weaving density [134]. PDS nanotextiles sutured into peritoneal wall of healthy mice preserved integrity during the experiment and demonstrated prolonged steady release and retention (56 days) of Ptx in peritoneum when compared to Taxol intraperitoneal injection. The ability of such stable implantable nanotextile to deliver chemotherapeutics at doses significantly lower than the maximum tolerated dose and the retention of medication in peritoneal organs as observed in the case of Ptx, could be exploited for treatment of advanced ovarian cancer which commonly metastases into peritoneal cavity.

Composite devices containing drug-loaded NFs are a research hotspot in regenerative medicine. For example, oriented core-shell NFs with dual drug load were used as filling for a new composite nerve guide conduit [135]. Electrical stimulation combined with VEGF and neuroprotective drug released from NFs promoted proliferation and differentiation of nerve cells. Similarly, nanofibrous tubes can be mounted on bare-metal stents as described in the case of diabetic artery disease therapy [136] or restenosis prevention [137].

Multilayered NF scaffolds are attractive multifunctional vehicles increasingly investigated for therapeutic applications. A recently prepared PLGA tri-layered scaffold was loaded with doxycycline, collagen and bupivacaine to promote healing, provide bone-mimicking scaffold and attenuate pain, respectively. Sustained release of both drugs was observed over 30 days and contributed to improved healing of ruptured tendon [138].

During the past three decades, several different nanoparticles, including polymer micelles, liposomes or solid nanoparticles, have been explored for precise delivery of therapeutic agents. One of the most explored administration routs is intravenous delivery. Although mentioned nanocarriers possess physico-chemical and synthetic plasticity which can be used for improvement of their stability under physiological conditions, protection of therapeutic load and functionalization, the exigency of existing barriers between the administration and target site can lead to significant loss of therapeutic cargo due to clearance, reduced accumulation in cells of interest or undesirable activation of DCs in the draining lymph nodes. Therefore, in some cases, localized administration emerged as a preferable delivery mode where NFs with controllable sustained release present suitable repositories able to complement positive aspects of drug-loaded nanosystems. In the case of micelle-in-NF loaded with small therapeutic drug, lower dosage of drug was needed to ensure therapeutic levels at tumor site than when intravenously administered micelles were used. Furthermore, minimal levels of systemic drug exposure were observed, and a reduced frequency of drug administration was required. Localized micelle release was complemented by targeting ligands which promoted micelle endocytosis [127]. The addition of nanostructures was also used to improve tensile and drug releasing properties of PVP-based NFs [128]. While PVP core was used to encapsulate unstable 5-fluorouracil (5FU), the addition of PCL and multi-walled carbon nanotubes (MWCNT) to the PVP shell resulted in improved sustainable drug release and mechanical properties. Careful consideration of core/shell composition and the amount of loaded drug resulted in promising post-surgical DDS for cervical cancer. Conjugation of levofloxacin to pSiNPs via cleavable thioester bond significantly decreased initial drug burst release from PCL NFs [131].

Apart from metal and polymer-based NPs and micelles, liposomes and solid lipid particles can also be incorporated into NFs. Blend electrospinning of PVA and liposomes based on 1,2-distearoyl-sn-glycero-3-phosphocholine/cholesterol (DSPC/Chol) or phosphatidylcholine (PC) gave rise to hybrid NFs, where chemical, thermal and mechanical stability of hydrophilic and biocompatible polymer was combined with high encapsulating efficacy of hydrophobic and hydrophilic drugs. Moreover, the polymeric NF scaffold can enhance poor stability and drug releasing properties of liposomes towards long-term. The inclusion of stabilizing Chol into liposomal formulations preserves stability of liposomes during electrospinning process and was reflected in high retention of encapsulated model hydrophilic drug, calcein [139]. The same methodology was used to prepare model NFs composed of PCL/gelatin blend NFs containing liposomes loaded with antibacterial and antioxidative epigallocatechin-3-gallate (EGCG) [140]. Glutaraldehyde crosslinking was used to increase gelatin component stability and to provide functional groups for immobilization of liposomes through chemical reaction between phospholipid amino groups and aldehyde carbonyls. Although, in this case, it was not clear whether electrospinning preserved spherical structures of liposomes or gave rise to phospholipid bilayers, good EGCG release and activity against UV- and H_2_O_2_-caused oxidative stress were achieved. Similarly, HIV-1 prophylactic formulations were prepared in electrospun PVA/liposome NFs loaded with tenofovir disoproxil fumarate (TDF) and emtricitabine (FTC) [132]. Good dispersibility of liposomes in hydrophilic PVA polymer led to better polymer alignment and decrease of NF diameter. The incorporation of liposomes also influenced the presentation of beads within NFs, characteristic of liposome agglomerates. When drugs are incorporated within the liposomes, the number of observed bulges decreases, presumably due to increased flexibility of liposomes caused by disturbed lipid packing. These morphological features were in stark contrast to those encountered in lipophilic PCL NFs loaded with TDF and FTC, where difference in polymer and drug polarities lead to disruption in crystalline domains and resulted in enlarged loaded NFs. Differences in polymer nature and manner of drug loading also influence biopharmaceutical properties of formulations, and mucoadhesiveness and drug release profiles were affected. Nonetheless, both NF systems can provide local microbicide protective concentrations of both drugs within the required short time frame after single vaginal administration in mice. Although both NF formulations were more efficient than the existing oral standard, higher liposome-in-NF performance make them interesting alternative to the existing prophylactic therapies.

### 4.2. Delivery of Peptides and Proteins

Nanofibers can be used as versatile platforms for the delivery of therapeutic proteins and peptides. They can offer much needed protection against proteolytic enzymes, deleterious compartmental pH or peptide aggregation which impair full in vivo therapeutic potential of peptides [141] while preserving structural and functional integrity. In this context, NFs can serve as carriers or a therapeutic component of nanoformulation (Table 3).

The low inflammatory nature of supramolecular peptide NFs and tunable immune response that often does not require the use of adjuvants makes them interesting platforms for vaccines and immunotherapies [149]. NFs constructed of β-sheet or α-helical fibrillizing domain and antigen epitope domain can elicit intrinsic immune responses [150]. For example, efficacy of NFs bearing CD8^+^ epitope induced a larger immune CD8^+^ T cell response in lung-draining lymph nodes when administered intranasally [142]. Compared to subcutaneously delivered NFs, more resident T cells were produced providing faster response to influenza infection. Chemical plasticity enables customized display of antigens of self-assembling NFs, as demonstrated in the model HIV vaccine where folded glycoprotein antigen valency was associated with increased magnitude and breadth of immune response [151].

Synchronization of growth hormone expression is one of the driving forces of bone regeneration. Early expression of connective tissue growth factor (CTGF) is responsible for migration and attachment of marrow stromal cells and angiogenesis, while steady expression of bone morphogenetic protein 2 (BMP2) promotes osteoblastogenesis and the formation of new bone. Inspired by these dynamics in the physiological bone healing process, the multilayered NF system was designed for controlled temporal release of BMP2 and CTGF [144]. BMP2 was incorporated in the PVA core during co-axial electrospinning and deposition of CTGF was accomplished by LbL process on top of silk fibroin (SF)/PCL shell. Negatively charged (SF/PCL)/PVA NFs were alternatively soaked in positively-charged chitosan and negatively-charged CTGF solutions (Figure 5).

LbL coating caused changes in NF mats and a coarser irregular surface was observed. At the same time, additional layers checked burst release of BMP2 while enabling fast CTGF release. These in vitro findings were corroborated in vivo using a bone regeneration animal model. Sustained BMP2 release was observed within a month after implantation, although BMP2 migrated to distant sites. Unlike BMP2, CTGF release reach a minimum after six days. Although the surrounding tissue replaced the scaffold within the time frame of the experiment, degradation of new vasculature was observed.

Apart from GF delivery, other functional peptides can be released in a sustained manner from NFs [146]. For example, anti-inflammatory neurotensin was used for diabetic wound healing [145]. In another study, elastic PCL membranes were conjugated to antibacterial bacteriophage capsids. Mobilized capsids demonstrated bactericidal activity immediately after the contact with *P. aeruginosa* and it was maintained even after 25 washing cycles [148]. The results suggest that this type of NFs could be used to prepare antimicrobial non-woven textiles and dressings.

High porosity enables mesoporous silicon nanoparticles (pSiNP) to efficiently encapsulate and protect various therapeutic biological molecules against enzymatic and nucleolytic degradation. Successive oxidation and hydrolysis to water-soluble Si acids imparts them with degradable biocompatibility and makes them attractive delivery systems for sensitive biomolecule. Additionally, they are able to protect the load against nonaqueous medium which broadens their applicability to NF production in non-aqueous solvents. pSiNPs loaded with the model protein lysozyme were produced by airbrush spray nebulization from chloroform solutions of PCL or poly(lactide-co-glycolide) (PLGA) [94]. Aligned hybrid NFs were efficient templates for directional growth of neurites and astrocytes and therefore hold the potential in neuronal regeneration therapies. Additionally, pSi photoluminescence in near IR (NIR) adds another dimension to multifunctional NFs and allows for in vivo imaging. Release of the model enzyme was sustained over the period of 30 days with no significant loss of function. Susceptibility to chemical modification of pSiNP extends loading options to various therapeutic molecules including small molecule drugs, RNA aptamers via calcium silicate condensation or proteins via an oxidative trapping technique [130,131]. Although the method was successfully applied in in vitro model of glioblastoma, its applicability in an in vivo model still needs to be tested.

### 4.3. Delivery of Polynucleotides

As in the case of proteins, the therapeutic potential of various polynucleotides (pDNA, mRNA and iRNA) is limited by poor stability under physiological conditions and potential immunogenicity, and NFs have emerged as a possible non-viral nanodelivery platform (Table 4).

Presence of cationic amino acids in polar heads of self-assembling peptide NFs can be exploited to form polyplexes with polynucleotides and can be used for cell transfection [156]. NFs composed of palmitoyl-GGGAAAKRKpeptide amphiphile can enter glioblastoma cells through endocytotic pathways and via passive translocation [152]. Gene silencing was successful after siRNA delivery and local injection into the glioblastoma site improved tumor load and increased survival.

The ability to mimic ECM became an attractive feature of NFs often explored for localized delivery of therapeutic molecules involved in regenerative medicine. One of the most used polymers for bone tissue engineering is PCL, which provides a convenient biomimicking scaffold of appropriate mechanical strength, slow degradability and sustained drug release. Apart from growth factors, polynucleotides such as silencing RNAs or DNA can be delivered by NFs [159,160]. miRNA polyplexes can be incorporated into NFs by electrospinning with polymer solutions. Although polyplex inclusion affects mechanical characteristics of NFs to some degree, increased hydrophilicity of loaded NFs promoted cell adsorption and miRNA-induced cell differentiation [157]. A similar approach was used for blood vessel regeneration [161]. Similarly, *N*-trimethyl chitosan chloride (TMC)-based pDNA polyplex was incorporated into composite PLA NFs. In this case, the addition of functional polyhedral oligomeric silsesquioxane (POSS) nanoparticles was used to increase the physicochemical properties of PLA matrix. The resulting PLA scaffold enabled sustained pDNA release over 35 days and efficiently promoted cell transfection [162]. Sustained pDNA release was also obtained by incorporating genetic material within core-shell NFs. Photothermal mats were used as carriers of pDNA/PEI polyplex which was packed into the NF core. To improve polyplex release, the PLA/gelatin shell was enriched with Au nanorods for photothermally induced pDNA release and cell membrane permeation [154].

In a recent study, the inclusion of photosensitive iron oxide NPs was used to achieve photoporation of a range of cells, including hard-to-transfect T-cells. Unlike in photoporation based on free light sensitive NPs where close contact of NPs and cells is necessary, in the case of hybrid NP-NFs, photoporation occurs without direct contact with cell membranes. Photoporation was successfully applied in cell transfection with biomolecules as small as siRNA and as large as CRISPR/Cas9 ribonucleoprotein complexes, albeit with less efficacy for large molecules [163]. Successful reduction of SKOV3 ovarian carcinoma tumor by chimeric antibody receptor T cells transfected with programmed cell death protein 1 (PD1) siRNA demonstrated the applicability of NP-NF for induced photoporation in cell engineering for therapeutic applications.

NFs with functionalized biomimetic coating were used as a platform for lipid nanoparticles (LNP) loaded with CRISPR/Cas9 ribonucleoprotein complex for acute myeloid leukemia therapy [158]. In this novel approach to CRISP/Cas9 delivery for leukemia stem cell (LSC) editing, PCL NFs coated with mesenchymal stem cell membranes (MSCM) mimic bone marrow matrix that is crucial for LSC survival. A negative MSCM surface immobilizes positively-charged LNPs via electrostatic interactions and is responsible for their sustained release. Finally, electrostatic incorporation of cytokine CXCL12α provides chemical gradient which drives LSC migration and recruitment onto MSCM-NFs (Figure 6).

NFs functionalized with MSCMs demonstrated improved loading of LNPs and chemokines and successful recruitment of LSCs. The ability to serve as an efficient drug reservoir improved LSC editing and significantly improved tumoral burden in orthotopic murine model injected with edited LSCs.

### 4.4. Cell Delivery

The ability of NFs to form ECM-mimicking structures is increasingly used to overcome translational challenges encountered in injection-based administration of cells in regenerative therapies and tissue engineering. Recent application of NF scaffolds in therapeutic cell delivery is summarized in Table 5.

Activation and mobilization of stem cells (SC) in the process of wound healing has identified them as a potential therapeutic strategy in wound treatment. The limited ability of SC homing coupled with low in vivo survival limits their therapeutic applicability and calls for efficient biocompatible delivery systems. The ability of β-chitin nanofibers (β-ChNF) to form hydrogel upon addition of cell culture medium was used to prepare ECM-mimicking scaffolds suitable for incorporation and sustained growth of adipose tissue-derived (AD) SC [164]. When applied in an in vivo model of wound healing, ADSC/ChNFs demonstrated improved epithelization, collagen deposition and angiogenesis which led to accelerated wound healing than was observed in control groups. Uniform distribution of ADSC within the porous three-dimensional structure of chitin hydrogels enabled sustained viability of ADSC and resulted in increased production of exosomes. The participation of exosomes in intercellular signal transduction combined with the secretion of paracrine factors were identified as a putative driving forces of wound healing. Similarly, hydrogels based on α-chitin nanofibers were able to participate in wound healing by serving as a 3D support for prolonged viability of bone marrow mesenchymal (BM) SCs. Additionally, they were able to induce BMSC differentiation into angiogenic cells and fibroblasts without addition of cell-differentiating factors [165]. Therefore, hydrogels based on natural polymers may present functional scaffolds with the promising future in clinical wound healing. Silk fibroid (SF) NFs seeded with human Wharton’s jelly MSC created artificial dermal layer that successfully interacted with the wound. Improving scaring was attributed to MSC differentiation into specialized skin cells and attenuation of cellular immune response [173]. Beneficial effects on wound healing were also observed when BMSC-derived keratinocytes seeded on chitosan/PVA/SF mats were applied [176].

A new approach of NF functionalization with cell-derived extracellular matrix (CDECM) resulted in formation of new functional biomimicking scaffolds. Silk nanofibrils coated with CDECM promoted proliferation of brown adipose (BA) stem cells into cardiac fibroblasts with the potential to mimic cardiac tissue specificity with future application in repair of myocardium damage [174].

Apart from NFs prepared from natural sources, scaffolds prepared from synthetic polymers also possess the ability to induce differentiation of MSC. Scaffolds containing PCL NFs coated with hydroxyapatite/tricalcium phosphate layer were prepared by fused deposition modeling technique and were tested for potential application as bone regeneration scaffolds [177]. PCL scaffold successfully provided long-term viability to several types of MSC: umbilical cord (UC), AD and BM. Although increased proliferation was observed for all MSC types after adhesion on the NF surface, the highest osteogenic potency was observed for BMSC [166]. Chitosan/PVA NF mats were recently used as a substrate for ADSC differentiation into chondrogenic cells indispensable for cartilage regeneration [175]. The fabrication technique can have significant influence on NF surface morphology and consequently on adsorbed cell function. Platelets seeded on centrifugally spun PCL NF released twice as much bioactive molecules than those grown on electrospun NFs. Under the influence of GFs produced by platelet-derivatized NFs, melanocyte growth was actively promoted and stimulated melanin synthesis which makes these cellular patches an interesting new therapy for vitiligo [172].

PLA is an attractive material for cell scaffolds used in bone regeneration due to its ability to promote the growth and differentiation of MSC to osteoblasts. However, this propensity seems to be reduced for PLA NFs spun from dichloromethane solutions. Addition of bone matrix component collagen to PLA NFs appears as a convenient strategy to improve the cell density of seeded cells and the osteoinductive property of PLA NFs. However, PLA-collagen blends resulted in NFs with insufficient pore diameter for successful scaffold colonization (<5 μm) which resulted in the absence of beneficial in vivo effects. To overcome these shortcomings, cell-loaded scaffolds were produced by direct incorporation of cells during NF fabrication. The combination of co-axial cell electrospraying during NF multi-jet electrospinning was optimized, with cell survival and the number of trapped cells strongly dependent on the collecting electrode type, spraying rate, spinning distance and number of spraying devices [170]. Although possible biological incompatibility between polymer composition (collagen) and used cell line (MSC) or organic solvent residues could be responsible for lower cell density and metabolic activity, combined electrospray/spinning process did not influence MSC differentiation into osteoblasts. Similar effects were observed in in vivo model where improved bone restoration was observed during the early phase of defect healing (four weeks) when compared to previously described scaffold models.

The ability of NFs to be processed into higher 3D structures can be used to prepare NF-based microspheres suitable for cell-based tissue engineering [171]. Unlike 3D scaffolds which need to be shaped and inserted into the desired site during surgical procedure, microspheres could be applied to irregular defects by minimally invasive techniques such as injection. Preparation of microspheres from already made NFs avoids the use of functionalized materials necessary for microsphere self-assembly while providing porosity inherent to NF scaffolds used for cell delivery. To prepare NF mats for microsphere formation, electrospun PCL-based NFs were segmented by cryo-cutting or sonication [167]. After that, fragmented NFs were electrosprayed into microspheres from aqueous suspensions. Obtained microspheres were successfully applied as growing scaffolds for rat bone-marrow-derived MSC and produced in situ microtissue-like structures. When murine embryonic SC were seeded, stimulation with retinoic acid induced formation of embryoid bodies containing neural progenitor cells. This novel microsphere design presents a potential for development of novel SC-based therapies of neurodegenerative diseases or treatment of ischemic locations. 

Injectable scaffolds can also be prepared from self-assembling peptide amphiphile (PA) NFs. Gels prepared by mixing lyotropic liquid crystals formed by PA NFs and skeletal muscle progenitor cells can assume architecture and stiffness of oriented muscle tissue [178]. In such a hierarchical structure, cells were distributed within and in between the PA NFs and have undergone alignment, proliferation and differentiation. Efficacy of this novel scaffold in muscle regeneration was confirmed in a murine in vivo model, where muscle repair by muscle stem cells was enhanced when PA NF liquid crystal gels were spiked with growing factors (vascular endothelial and basic fibroblast growth factors). While injectable NF formulations for cell delivery are based on more complex structures, injectable administration of conventional 3D NFs is possible in the case of core-shell NFs. However, recent studies emphasize the need of lubricant gel-like shell to produce what was termed SLIDING NFs (slidable injectable and gel-like) NFs [169,179]. Injectability of such fibers depends on fiber mobility, and fiber re-arrangement during injection changed mechanical properties of SLIDING fibers producing stiffer and oriented fibers, but did not affect the number, viability or proliferation of seeded cells. When applied in an animal model of human ischemic stroke by intracerebral injection through a catheter, NFs preserved survival of neural SC by preventing infiltration of inflammatory cells.

## 5. Final Remarks and Future Perspectives

The availability of biocompatible and biodegradable natural and synthetic polymers that can be processed into nanofibers offers enormous potential for the design of solid and injectable nanocarriers for chemo and biotherapeutics. The attractiveness of local delivery offered by nanofibers presents the opportunity to mitigate limited bioavailability and possible off-target side-effects that are often an issue with other types of systemically applied therapeutic nanocarriers. Polymer blends or composites obtained by the addition of other materials are widely used to improve mechanical, physical and biological properties. Considering the diversity of existing fabrication techniques, available polymers and used therapeutics, a complex interplay or parameters demand a case-by-case optimization to control shape, porosity and degradability, and therefore drug release. In polymer nanofibers used in regenerative medicine, adjustment of these parameters enables mimicking of mechanical and functional properties of tissues, and physiological templates for cell adherence, proliferation and differentiation. Although nanofibers have shown incredible potential for drug/biomedicine delivery applications there are still issues that should be resolved before nanofibers can be utilized in clinical practice. As a result, only eight clinical trials have been registered using the term “nanofiber” in the clicaltrial.gov database, based on a search performed on the 27 June 2022 (Table 6).

To date, specific central points related to the design of functional nanofibers, including optimization of processing parameters, drug loading and controllable release relevant for achieving desired therapeutic outcomes seem to be at the forefront of the majority of published papers in the area, and the majority of biological tests is conducted in vitro, especially when nanofibers are applied in the field of cancer therapy. In the last decade, adaptability of electrospinning to produce NFs of various morphologies and hierarchical 2D and 3D structures, porosities and surface properties made it the most exploited small-scale producing method. Relatively simple post-production modification and incorporation of diverse therapeutic molecules that resulted in controllable release profiles made it the source of the most studied NFs. Despite of progress that is made in scaling of electrospinning necessary for industrial production and clinical translation, other simpler methods are emerging on the therapeutic nanofiber scene. However, their presence in scientific literature in still lagging behind the well represented and described electrospinning. Therefore, the full potential of solution spinning or hybrid methods which are attracting increased attention in the last decade is still to be discovered. The number of existing and emerging technologies can be optimized to obtain NFs of desired mechanical characteristics and drug loading/releasing properties. However, the wide choice of available polymers and therapeutic agents requires individual approach to their optimization, and the lack of comparative studies and detailed insight into the most recent methods, makes it difficult to single out the ideal or most appropriate production technique in regard to mechanical properties or loading/release of different therapeutic agents. Finally, the selection of the method of choice should also be dictated by their compatibility with the polymers and the therapeutic agent, administration mode as well as the interactions of the resulting NF with the biological system.

Although most of the polymer-based nanofibers seem to be well-tolerated, certain additives may raise toxicological and environmental concerns [180,181]. For example, materials with good electrical conductivity such as graphene are used in polymer mats for cardiac tissue regeneration. In the case of graphene, observed toxicity to cardiac cells and animals called for extensive in vitro and in vivo toxicological assessment of nanofibers in relation to therapeutic concentration and time [182]. Although generally considered as safe biocompatible material, recent toxicological study of cellulose nanofibers revealed that the short-tern exposure can be cytotoxic to skin and fibroblast cells but without eye and skin irritation [183]. However, this study suggests that long-term studies should also be considered, and should be a part of good practice for nanofibers with potential biomedical application [184,185]. Other, technology-related challenges also remain to be resolved. For example, the impact of functionalization methods and residual solvents on functionality and in vitro and in vivo compatibility of nanofibers should be considered. Reproducibility of physico-chemical properties, fiber orientation or surface properties is indispensable for the use of NF in tissue regeneration. Despite of exciting potential that nanofibers display in drug/biomedicine delivery within the academic studies, there is a crucial need for efficient industrial scale-up of production methods before successful transfer into widespread clinical practice.

## Figures and Tables

**Figure 1 ijms-23-08581-f001:**
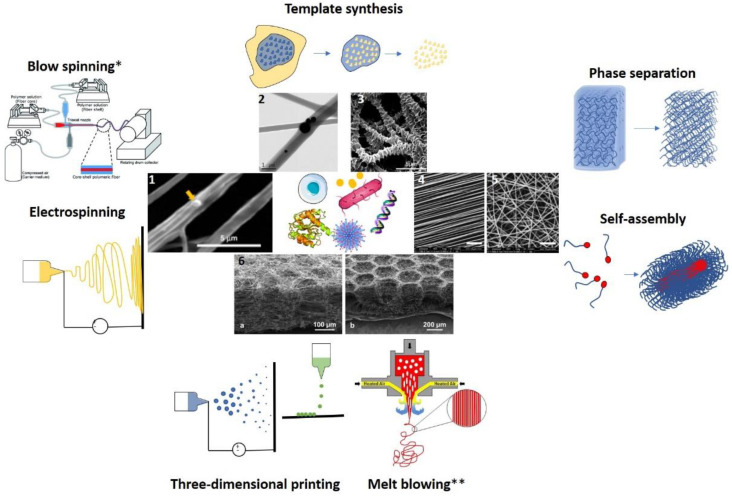
The overview of techniques for nanofiber fabrication. Some of the many possible nanofiber morphologies are presented in the central part: (**1**) Core-shell NFs obtained by microsol electrospinning loaded with liposomes (yellow arrow). Adapted from [9]. Copyright © 2020, The Author(s) under CCBY licence. (**2**) Janus NFs loaded with AgNP and ciprofloxacin prepared by side-by-side electrospinning. Reprinted from [10] with permission from Elsevier. (**3**) Hierarchical shish-kebab core-shell NF prepared for growth factor delivery. Adapted with permission from [11]. Copyright 2020 *American Chemical Society*. (**4**) and (**5**) Aligned and random oriented NFs. Adapted from [12] under CC BY 4.0. (**6**) 3D nanofibrous construct prepared by electrospinning and electrospraying. Reprinted from [13], with permission from Elsevier. In the innermost part, therapeutic agents that can be delivered by nanofibers are presented. * Reprinted with permission from [14] the Royal Society of Chemistry. ** Reprinted with permission from [15]. Copyright 2013, American Chemical Society.

**Figure 2 ijms-23-08581-f002:**
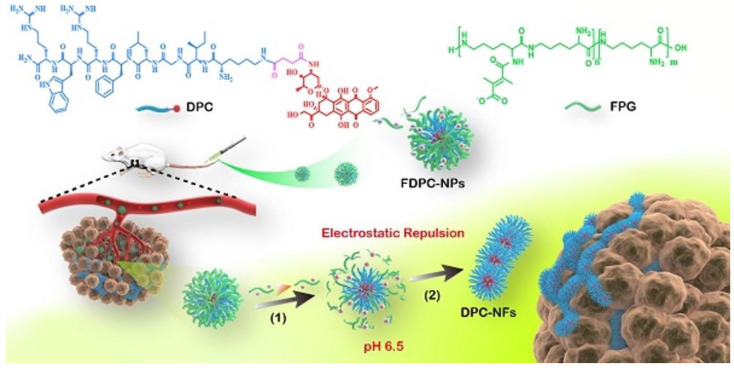
The self-assembly of Dox-polypeptide conjugates (FDPC-NPs) and the morphologic transformation of the acid-responsive FDPC-NPs in vitro and in vivo. Change in tumor site pH causes disintegration of micelle shell (1) and subsequent assembly of nanofibers (2) driven by π–π stacking, leading to long-term drug retention in the tumor [27]. Reprinted with permission from [27], Copyright 2022, Ivyspring International Publisher.

**Figure 3 ijms-23-08581-f003:**
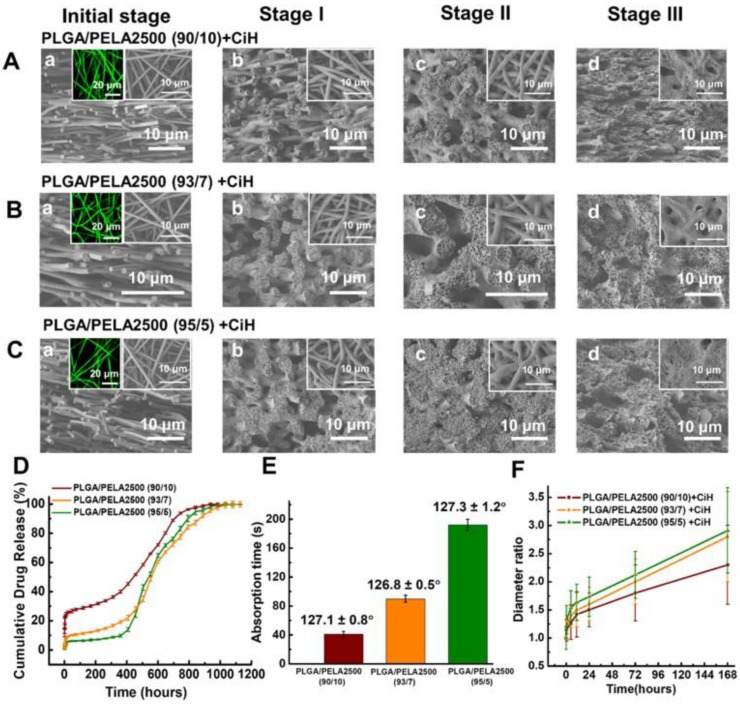
Characterization of PLGA/PELA2500/CiH electrospun membranes with different contents of the second component PELA2500. (**A**–**C**) Cross-sections of PLGA/PELA2500/CiH membranes: (**A**) PLGA/PELA2500 (90/10); (**B**) PLGA/PELA2500 (93/7); (**C**) PLGA/PELA2500 (95/5). (**D**) In vitro drug release profiles of PLGA/PELA2500/CiH membranes. (**E**) Changes in the absorption time of PLGA/PELA2500/CiH membranes the values of the contact angles of the membranes. (**F**) Changes in the fiber diameter ratio of PLGA/PELA2500/CiH membranes. © 2020 The Authors. Published by Elsevier B.V, under CC-BY-NC-ND license.

**Figure 4 ijms-23-08581-f004:**
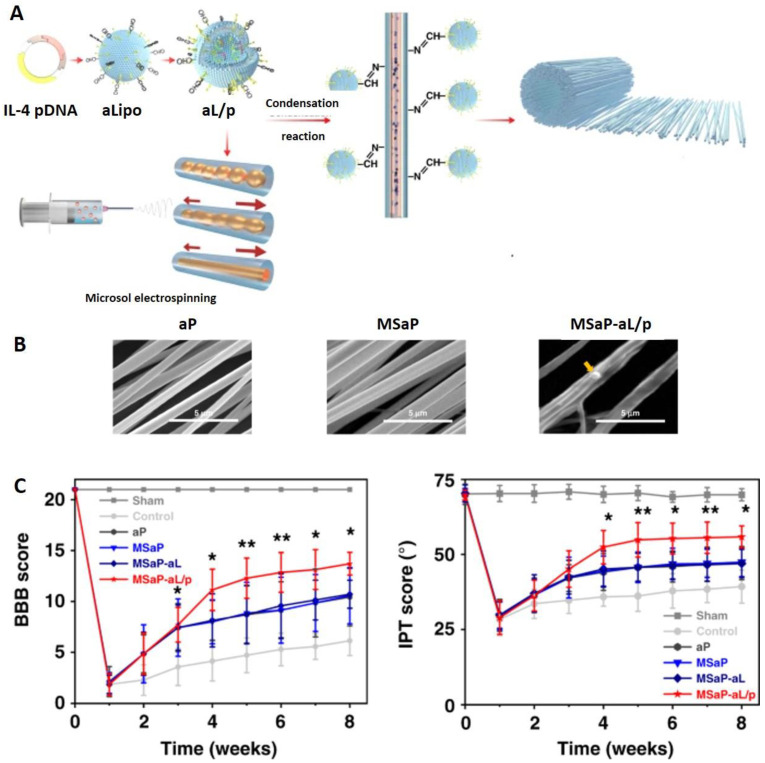
Combination neuron therapy by NF: (**A**) Construction of oriented fiber scaffold loaded with nerve growth factor by microsol electrospinning is followed by condensation with pDNA-loaded liposomes. (**B**) Morphology of different NF scaffolds (aP: amino PLA NF; MSaP: aPLA microsol fibers; MSaP-aL/p: MSaP conjugated with pDNA-loaded liposomes where the yellow arrow indicates the position of liposome). (**C**) Evaluation of motor function recovery by BBB and IPT scores. * *p* < 0.5, ** *p* < 0.01. Adapted from [9]. Copyright © 2020, The Author(s) under CCBY licence.

**Figure 5 ijms-23-08581-f005:**
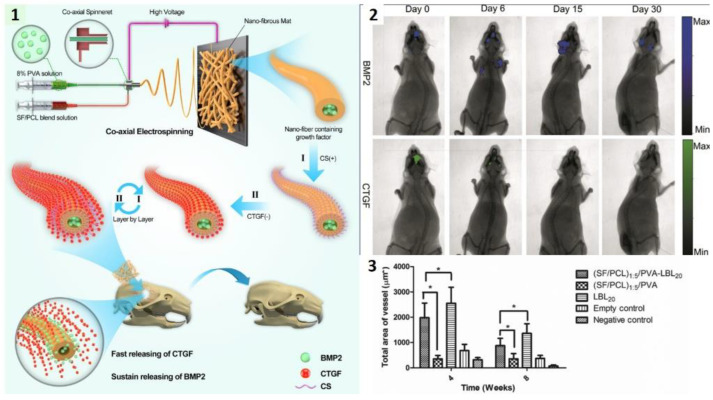
Multilayer NF loaded with growth factors for bone healing ((SF/PCL)_1:5_/PVA-LBL20). (**1**) Core-shell NFs with MBP2-loaded core is functionalized by successive layers of chitosan and the second growth factor, CTGF. (**2**) Time-controlled release of growth factors after intracranial implantation. (**3**) Temporal development of microvasculature formation. * *p* < 0.5 Adapted with permission from [144]. Copyright 2019 American Chemical Society.

**Figure 6 ijms-23-08581-f006:**
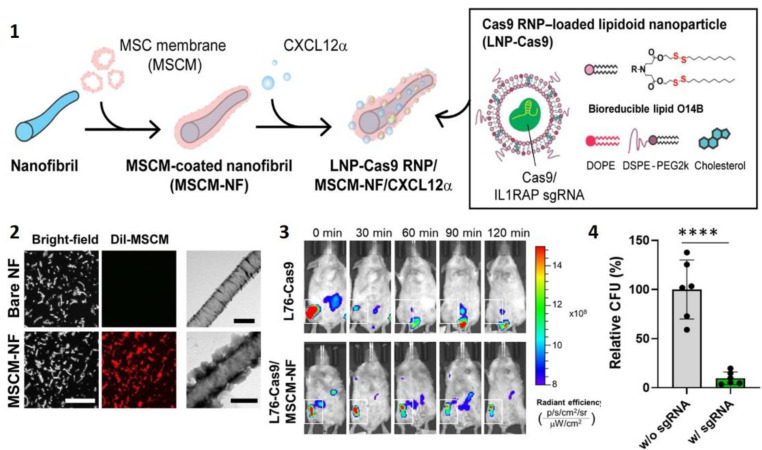
Injectable biomimetic nanofiber formulation for acute myeloid leukemia treatment: (**1**) PCL nanofibril coated with mesenchymal stem cell membrane (MSCM) loaded with cytokine CXCL12α and lipid nanoparticles-CRISPR/Cas9 (LNP-Cas9) complex. (**2**) Images of naked NF and MSCM-coated NF. (**3**) Biodistribution of MSCM-NF/LNP-Cas9 after injection into the bone marrow of right tibia indicates prolonged localization in tibia when compared with LNP-Cas9. (**4**) Leukemia stem cells exhibit reduced colony formation after treatment with therapeutic MSCM-NF/LNP-Cas9. **** *p* < 0.0001. Adapted from [158]. Copyright © 2021, The Author(s) under CCBY 4.0 license.

**Table 1 ijms-23-08581-t001:** Overview of the external stimuli used in design of smart nanofibers.

Polymer	Solvent	Therapeutic Agent	Fabrication	Fiber Diameter	Architecture	Therapeutic Outcome	Ref
**Thermo-responsive Nanofibers**							
Eudragit^®^ RS 100 PMMA	DMF	Octenidine	Single nozzle electrospinning	134–168 nm	Single mesh	Reduced in vitro bacterial colony formation	[104]
PCL/PNIPAAM	TFE	Doxycycline hyclate	Single nozzle electrospinning followed by mesh immersion and UV crosslinking	275–490 nm	Core-crosslinked shell mesh	Reduced in vitro bacterial colony formation	[106]
P(NIPAAm-*co*-NIPMAAm)/PLCL	H_2_O CH_2_Cl_2_/DMF		Light-assisted co-axial electrospinning	1400 nm	Crosslinked core-shell	Enhanced drug penetration in soft tissues by NIR-light assistance	[111]
poly(NIPAAM-co-AAh)/PEGDMA	Ethanol TFE	Bovine serum albumin Dexamethasone phosphate	EHD side-by-side co-jetting followed by UV crosslinking	~1000 nm	Bicompartmental nanofibers	-	[112]
PGS/PCL	CH_2_Cl_2_/ Ethanol	NP-encapsulated cefazolin and ceftriaxone	Single nozzle electrospinning followed by printing of electrically conductive pattern	350–1100 nm	Elastic sheets	Reduced in vitro bacterial colony formation	[113]
**pH-responsive Nanofibers**							
PVA/p(4VP-*co*-EGDMA)	H_2_O	Rose Bengal	Single nozzle electrospinning followed by chemical vapor deposition	580 nm	Core-coated mats	In vitro antiproliferative activity against cancer cells (U87MG)	[114]
aPLA/HA	CH_2_Cl_2_/DMF H_2_O	Interleukin-4 pDNA-loaded liposome Nerve growth factor	Microsol electrospinning followed by chemical grafting of liposomes	550–570 nm	Oriented core-shell fiber scaffolds	Reduced in vivo inflammatory response, increased nerve repair and recovery of motor function	[9]
**Redox-responsive Nanofibers**							
PEO/PCL and redox responsive c-6A PEG-PCL/6A PEG-PCLSH NG	CH_2_Cl_2_ H_2_O	BMP-2	Co-axial electrospinning	220–340 nm	Core-shell	In vitro stem cell osteogenesis differentiation. In vivo mandible bone reconstruction	[115]
**Enzyme-responsive Nanofibers**							
coPEA 8-[L-Phe-6] 0.95–[L-Leu-6]0.05/ PCL	H_2_O CH_2_Cl_2_/C_6_H_12_	a-chymotrypsin Non-specific lipase Fluorophore	Colloid electrospinning	375–1380 nm	Single scaffolds	-	[93]
HA-SH PEO	H_2_O	Tenofovir	Co-axial electrospinning	75 nm	Core-shell scaffolds	In vitro anti-HIV effect	[116]
P18-PLGVRGRGD	H_2_O	P18	Gelatinase-triggered in situ formation	30 nm	Fibrous superstructures	Improved photoacoustic tumor imaging and therapeutic efficacy	[30]
**Light-responsive Nanofibers**							
PNIPAM/ Au nanorods	DMF/THF	Camptothecin	Single nozzle electrospinning followed by high temperature cross-linking	600–700 nm	Single scaffolds	In vitro decreased viability of malignant glioma cells (U-87 MG)	[117]
PVA/PVP-FeOOH	H_2_O	Methylene blue	Co-axial electrospinning	580 nm	Core-shell mat	-	[118]
PLLA/P(NIPAAm-co-PNIPMAAm) AuNR hydrogel	CHCl_3_/DMF	Rhodamine B	Single nozzle electrospinning followed by pillow assembly and interface UV-crosslinking		Nanostructured pillow composed of NF membrane and hydrogel core		[111]
**Magneto-responsive Nanofibers**							
PVA/IONP	H_2_O	Acetaminophen	Infusion gyration followed by drug adsorption	100−300 nm	Single mats of beaded fibers	-	[119]
**Mechano-responsive Nanofibers**							
P(VDF-TrFE)	DMF/THF	Crystal violet poly(L-lysine)-Vivotag-645	Single nozzle electrospinning	70–500 nm	Nanofibrous membranes	-	[120]
**Multi-responsive Nanofibers**							
PNVCL/EC Eudragit L100	Ethanol	Ketoprofen	Twin-jet electrospinning	350–670 nm	Thermo- and pH-sensitive hybrid mats	-	[110]
PLGA/ IONP	HFIP	Bortezomib	Single nozzle electrospinning followed by functionalization	600 nm	Not-like nanofibers	In vitro apoptosis in breast cancer cells (4T1)	[121]

Abbreviations: PNIPAM: Poly(N-isopropylacrylamide); PCL: polycaprolactone; TFE: 2,2,2 trifluoroethanol; NVCL: N-vinylcaprolactam; MAA: methacrylic acid; DMAC: N,N-Dimethylacetamide; EHD: electrohydrodynamic; poly(NIPAM-co-Aah): poly(N-isopropylacrylamide-co-allylamine hydrochloride); PEGDMA: poly(ethylene glycol) dimethacrylate; BMP-2: bone morphogenetic protein 2; PEO: Polyethylene oxide; c-6A PEG-PCL/6A PEG-PCLSH NG: polyethyleneglycol-polycaprolactone/6 arm polyethyleneglycol-polycaprolactone-sulfhydryl nanogel; P(NIPAAm-*co*-NIPMAAm): poly(*N*-isopropylacrylamide-*co*-*N*-isopropylmethacrylamide); PLCL: poly-l-lactide-*co*-caprolactone; DMF: dimethylformamide; P(VDF-TrFE): poly(vinylidene fluoride-trifluroethylene); PVA: polyvinyl alcohol; p(4VP-*co*-EGDMA): poly(4-vinylpyridine-*co*-ethylene glycol dimethacrylate); PNVCL: poly(N-vinylcaprolactam); EC: ethyl cellulose; coPEA: copoly(ester amide); C_6_H_12_: cyclohexane; aPLA: amino polylactic acid; HA: sodium hyaluronate; HA-SH: thiolated HA; PVP: polyvinylpyrrolidone; PLGA: poly(D,L-lactide-co-glycolide); HFIP: 1,1,1,3,3,3-hexafluoro-2-propanol; NP: nanoparticle.

**Table 2 ijms-23-08581-t002:** Nanofiber systems used for delivery of small therapeutic drugs.

Polymer	Therapeutic Agent	Functionalization	Fabrication	Architecture	Administration Route	Therapeutic Outcome	Ref
Amphiphilic peptides	Liver X receptor agonist GW3965	ApoA1-targeting peptide	Self-assembly	Individual NFs with secondary β-sheet structure	Intravenous	Reduced plaque burden	[123]
Amphiphilic peptides	Taxol	FGL	Self-assembly	NFs hydrogel	Intraspinal microinjection post-SCI	In vitro and in vivo neurite elongation. Decreased inflammatory response and neurobehavioral recovery in vivo	[124]
PLA	Paclitaxel	-	Single nozzle electrospinning	NF membrane	-	Inhibition of cancer cells growth (HCT-116)	[125]
PCL	Lipophosphonoxin	-	Single nozzle electrospinning	NF scaffold	NF wound dressing	Reduced *Staphylococcus aureus* count in infected wound. No systemic absorption	[126]
PVA/ crosslinked gelatin	Dox-loaded PCL-PEG micelle	Folate-decorated micelles	Coaxial electrospinning	Core-shell NFs	Localized implant	In vitro and in vivo antitumor effect against mammary tumor	[127]
PCL/PVP	Multi-walled carbon nanotubes 5-fluorouracil	-	Coaxial electrospinning	Core-shell NF	-	In vitro cytotoxicity against HeLa cervical tumor cells	[128]
PLA/PLGA	Aceclofenac or insulin	-	Core electrospinning followed by sheet electrospinning	Twisted core yarn coated with drug-loaded sheet	Surgical sutures	Sustained drug release attenuated skin inflammation in in vivo animal model	[129]
PCL	pSiNP/drug, enzyme or RNA aptamer pSiNP/levofloxacin	- -	Airbrush spray nebulization. Single nozzle electrospinning	Aligned NFs NF mat	- -	Directional growth of neuronal cells	[94,130,131]
PVA	Liposomes loaded with tenofovir disoproxil fumarate and emtricitabine	-	Single nozzle electrospinning	NF mat	-		[132]

Abbreviations: CDN: cyclic-dinucleotide; PDA: polydiacetylenic; PAA: poly(acrylic acid); rGO: reduced graphene oxide; POSS: functional polyhedral oligomeric silsesquioxane; PVA: Poly-(vinyl alcohol); FGL: neural cell adhesion molecule motive; SCI: spinal cord injury.

**Table 3 ijms-23-08581-t003:** Nanofiber systems used for the delivery of therapeutic protein and peptides.

Polymer	Therapeutic Agent	Fabrication	Architecture	Administration Route	Therapeutic Outcome	Ref
Q11 self-assembly domain	Influenza acid polymerase epitope	Self-assembly	Individual NFs	Intranasal Subcutaneous	Reduced viral load in the lungs 6 weeks after 2nd vaccination	[142]
PEG-Q11R9	CDN mucosal adjuvants	Self-assembly	Individual NFs	Sublingual	Activation of dendritic cells in the draining lymph nodes	[143]
SF/PCL/PVA	BMP2 CTGF	Coaxial electrospinning followed by layer-by-layer deposition	Multilayer core-shell NF	Implant	In vitro and in vivo temporal control of GF release Promoted bona and vessel formation	[144]
PLGL/CNC	Neurotensin	Single nozzle electrospinning	Composite NF membrane	Wound patch	Accelerated skin regeneration	[145]
PCL	Pexiganan	Single nozzle electrospinning followed by hydrolysis	NF mats	-	In vitro antimicrobial activity on gram-positive and negative bacteria	[146]
PAA/rGO β-cyclodextrin	Insulin	Single nozzle electrospinning	Hydrogel-like fiber mats	Buccal Eye cornea	Ex vivo delivery via eye cornea and the buccal mouth lining	[147]
PCL	Bacteriophage capsid	Single nozzle electrospinning followed by bacteriophage conjugation	Elastic NF mat	-	In vitro bactericidal activity (*P. aeruginosa*)	[148]

Abbreviations: CDN: cyclic-dinucleotide; PAA: poly(acrylic acid); rGO: reduced graphene oxide; SF: BMP2: bone morphogenetic protein 2; CTGF: connective tissue growth factor; SK: silk fibroin; CNC: cellulose nanocrystals.

**Table 4 ijms-23-08581-t004:** Nanofiber systems used for the delivery of polynucleotides.

Polymer	Therapeutic Agent	Functionalization	Fabrication	Architecture	Administration Route	Therapeutic Outcome	Ref
palmitoyl-GGGAAAKRK peptide amphiphile	siRNA	-	Self-assembly	Individual NFs	Intra-tumoral administration	Extended survival of glioblastoma bearing mice	[152]
PDA	siRNA	-	Photopolymerization	Individual NFs	Intraperitoneal	In vitro and in vivo transfection of cancer cells and oncogene silencing	[153]
pDNA-PEI PLA-gelatin	pDNA polyplex	Au nanorods	Coaxial electrospinning	Nanofiber mats	-	Improved in vitro transfection	[154]
PVA	pDNA-CHAT polyplex	Cell penetrating CHAT peptide	Single nozzle electrospinning followed by DNAsoak-loading	Crosslinked nanofibers	-	Improved in vitro transfection	[155]
Virus-inspired polypeptide gene vector capped with lipophilic tail	pDNA	tetraphenylethene	Self-assembly	Individual NFs	-	In vitro transfection of wide cell range, including stem cells	[156]
PCL/gelatin	miRNA polyplex	-	Dual power electrospinning	Bilayer electrospun membranes	*-*	Improved in vitro osteogenic differentiation of human-induced pluripotent stem cells	[157]
PCL	LNPs loaded with CRISPR-Cas9 RNP complex	Biomimetic MSCM coating CXCL12 chimokine	Single nozzle electrospinning	Fragmented nanofibrils with biomimetic coating	Intra bone-marrow injection	Reduced human leukemic burden in mice	[158]
**Combination therapy**							
HA/aPLA	pDNA-loaded aLiposome nerve GF	-	Microsol Electrospinning followed by chemical grafting of pDNA-aLiposome	Oriented core-shell NF	Implantation	Reduced in vivo inflammatory response, increased nerve repair and recovery of motor function	[9]

Abbreviations: CDN: cyclic-dinucleotide; PDA: Polydiacetylenic; PAA: poly(acrylic acid); rGO: reduced graphene oxide; POSS: functional polyhedral oligomeric silsesquioxane; PVA: Poly-(vinyl alcohol); GF: growth factor; aPLA: aminated PLA.

**Table 5 ijms-23-08581-t005:** Nanofiber systems used for delivery of polynucleotides.

Polymer	Therapeutic Cell	Fabrication	Architecture	Therapeutic Outcome	Ref
β-chitin	ADSC	Chemically from natural sources	NF hydrogel	Improved in vivo wound healing	[164]
α-chitin	BMSC	Chemically from natural sources	NF hydrogel	Improved in vivo wound healing	[165]
PCL	BMSC, ADSC, UCSC	Fused deposition modeling	NF directed scaffolds	Enhanced in vitro proliferation and osteogenic potency	[166]
PCA/gelatin PLGA/gelatin	MSC, ESC	Electrospinning followed by electrospraying	Injectable NF microspheres	In vitro neural differentiation of ESC	[167]
Amphiphilic peptides	MuSC, VEGF, bFGF	Self-assembly followed by annealing and gelation	Injectable liquid crystalline NF gel	Enhanced engraftment of transplanted MuSCs	[168]
PCL-PS/MeHA	hNSC	Axial electrospinning followed by gel coating and crosslinking	SLIDING NFs	Prolonged survival of cells	[169]
PLA/collagen	MSC	Simultaneous co-axial cell electrospraying and polymer electrospinning	Cell-loaded NFs	Enhanced bone regeneration	[170]
Chitosan/PVA/SF	BMSC-derived keratinocytes	Single nozzle electrospinning followed by crosslinking	NF mat	Improved wound healing and skin regeneration	
PLA	ADSC	Single nozzle electrospinning followed by fragmentation and mineralizarion	Composite spheroids based on mineralized NF	Sustained ADSC viability and enhanced expression of osteogenic genes	[171]
PCL	Melanocyte	Single nozzle electrospinning followed by platelet soaking	Functionalized NF mats	Sustained melanocyte growth under the influence of GFs secreted by platelets Stimulated melanin synthesis for vitiligo treatment	[172]
SF	Wharton’s jelly MSC	Single nozzle electrospinning	Crosslinked NF 3D mat	Improved in vivo wound healing	[173]
SF	BASC	Salt sinking followed by CF growth and decullarization	ECM-coated NF mat	In vitro differentiation of BASC into CF	[174]
Collagen/PVA	ADSC	Single nozzle electrospinning	NF 3D mat	In vitro differentiation of ADSC into chondrogenic cells	[175]

Abbreviations: SC: stem cell; AD: adipose derived; BM: bone marrow; UC: umbilical cord; MSC: mesenchymal SC; ESC: embryonal SC; Mu: muscle; VEGF: vascular endothelial growth factor; bFGF: basic fibroblast growth factor; PLCL: Poly(lactide-co-ε-caprolactone); Gel-MA: gelatin-methacrylate; Me-HA: methylacrylated hyaluronic acid; hNSC: human neural SC; BASC: brown adipose SC; CF: cardiac fibroblasts; ECM: extracellular matrix; CF: cardiac fibroblasts; SF: silk fibroin.

**Table 6 ijms-23-08581-t006:** A summary of the registered clinical trials using the term “nanofiber.” Data was collected from the clicaltrial.gov database based on a search performed on the 27 June 2022.

NCT Number	Phase	Status	Condition(s)	Intervention(s)
NCT 04325789	Not Applicable	Recruiting	Rotator Cuff Tears	Nanofiber scaffold
NCT 03690960	Not Applicable	Unknown	Necrosis, Pulp	Electrospun TAP nanofibers
NCT 03242291	Not Applicable	Unknown	Marginal Integrity of Hydroxyapatite Nanofiber Reinforced Flowable Composite	Conventional resin-based flowable composite. Hydroxyapatite Nanofiber reinforced flowable composite
NCT 03264105	Not Applicable	Unknown	Retention Rate of Flowable Composite in Demineralized Pits and Fissure	Conventional resin-based flowable composite. Hydroxyapatite Nanofiber reinforced flowable composite
NCT 04867265	Not Applicable	Completed	Ventilation During Resuscitation	Mouth-to-mouth ventilation
NCT 04870736		Withdrawn	Basic Life Support Ventilation	Mouth-to-mouth ventilation
NCT 02237287	Early Phase 1	Terminated	Wounds|Pressure Ulcer	Wound dressing with VAC and sNAG without anti-aggregation
NCT 02680106	Not Applicable	Completed	Wound of Skin	Experimental: SPINNER. Active Comparator: JELONET/IBU Biatain

Abbreviations: IBU: Biatain dressing used for split-skin donor-site wounds; JELONET: gauze dressing used for split-skin donor-site wounds; sNAG: poly-N-acetyl glucosamine nanofibers; SPINNER: a hand-held, portable electro-spinning device that produces personalized in-situ nanofiber dressings for the treatment of external burns and wounds; TAP: a triple antibiotic mixture consisting of ciprofloxacin (CIP), metronidazole (MET) and minocycline (MINO); VAC: vacuum-assisted closure.

## Data Availability

Not applicable.

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
