# Peer review of "Nanofiber Carriers of Therapeutic Load: Current Trends"

_ijms, 2022, doi:10.3390/ijms23158581_

Round 1

Reviewer 1 Report

This is a very interesting and comprehensive review article. It was a pleasure to review it. I have only one wish for the authors: It is necessary to improve the quality of figures 1 and 4

Author Response

Manuscript ID: ijms-1819881

Title: Nanofiber carriers of therapeutic load: Current trends

Dear Editor,

Doctor Brian Chen

The authors appreciate the careful review and greatly acknowledge the comments that the Reviewer has provided on our manuscript. We have carefully revised the manuscript and have made the recommended changes and answered in detail to the questions raised. Additional information was also added when appropriate. All changes made to the text are highlighted in yellow colour and marked up using the “Track Changes” function.

# Reviewer 1

This is a very interesting and comprehensive review article. It was a pleasure to review it. I have only one wish for the authors: It is necessary to improve the quality of figures 1 and 4

Response:

 We thank the reviewer for suggestion and have improved the figure quality.

Reviewer 2 Report

The article written by Jarakv et al. provides a comprehensive overview of nanofibers as carriers for drug delivery applications. The review outline advances in nanofiber-based delivery systems,  methods, and their potential biomedical applications. Overall, the review is well written and scientifically sound, a few additions are essential to improve the impact of this article. 

Major

1)    Section 2 details various methods of fabricating nanofibers. However, emphasis has been given only to ‘electrospinning’ and its parameters. Details for other methods should be provided or a justification for detailing electrospinning should be explained.

2)    Section 3.-Drug loading and release- This method details only the approaches based on electrospun fibers. Authors should either include other methods to generalize the discussion or should restructure the review centered on electrospinning.

3)    Authors should discuss the efficiency of drug loading for different methods and provide an understanding of the most appropriate method to deliver small molecules and biologics.

4)    The review is not focused on recent advancements, for instance, drug delivery based on core-sheath electrospun yarns, and its applications have to be included. 

Electrospun polymeric core-sheath yarns as drug eluting surgical sutures

Minor 

1)    Line 14: Rewrite this sentence. Nanofibers are solid fibres that have a diameter under a micrometre. Replace ‘solid’ with polymeric/composite and ‘micrometre’ with nanoscale

2)    Elaborate the section 2.1,  2.2, 2.4 and 2.5

3)    Section 3, titled ‘Drug loading and release’ should be changed to ‘Delivery of therapeutics’

4)    Include recent citations. 

Author Response

Manuscript ID: ijms-1819881

Title: Nanofiber carriers of therapeutic load: Current trends

Dear Editor,

Doctor Brian Chen

The authors appreciate the careful review and greatly acknowledge the comments that the Reviewer has provided on our manuscript. We have carefully revised the manuscript and have made the recommended changes and answered in detail to the questions raised. Additional information was also added when appropriate. All changes made to the text are highlighted in yellow colour and marked up using the “Track Changes” function.

# Reviewer 2

The article written by Jarakv et al. provides a comprehensive overview of nanofibers as carriers for drug delivery applications. The review outline advances in nanofiber-based delivery systems, methods, and their potential biomedical applications. Overall, the review is well written and scientifically sound, a few additions are essential to improve the impact of this article. 

Major

1) Section 2 details various methods of fabricating nanofibers. However, emphasis has been given only to ‘electrospinning’ and its parameters. Details for other methods should be provided or a justification for detailing electrospinning should be explained.

Response:

We thank the Reviewer for the comment. We have included justification for singling out electrospinning in the preface to the Section 2. That section reads:

The area of therapeutic NF production is an extremely fertile filed of nanomedicine. Despite of variety of methods that have been developed so far (Figure 1), during the last decade or so, some methods have become more represented in scientific research. The broad applicability in regard to polymer material and therapeutic molecules, controllable morphologies and therapeutic load release, or simplicity of use are some of the factors that made those methods highly attractive from nanotherapeutic point of view. In this chapter these methods will be described in greater detail. Due to the prevalence of electrospinning in NF fabrication, special attention will be paid to this production method.

 We have also added more details to the descriptions of some other methods, namely Blow spinning and 3D printing due to their representation in the scientific reports during the last decade. Changes are labelled in yellow.

2) Section 3. Drug loading and release- This method details only the approaches based on electrospun fibers. Authors should either include other methods to generalize the discussion or should restructure the review centered on electrospinning.

Response:

We thank the Reviewer on the comment. We have generalized the discussion by commenting other methods as well. However, since electrospinning is the most investigated method in the last decade, detailed information on these issues is still scarce. The additions in Chapter 3 are labelled in yellow.

3) Authors should discuss the efficiency of drug loading for different methods and provide an understanding of the most appropriate method to deliver small molecules and biologics.

Response:

In the last decade, adaptability of electrospinning to produce NFs of various morphologies and hierarchical 2D and 3D structures, porosities, surface properties made it the most exploited small-scale producing method. Relatively simple post-production modification and incorporation of diverse therapeutic molecules that resulted in controllable release profiles made it the source of the most studied NFs. Despite of progress that is made in scaling of electrospinning necessary for industrial production and clinical translation, other simpler methods are emerging on the therapeutic nanofiber scene. However, their presence in scientific literature in still lagging behind the well represented and described electrospinning. Therefore, the full potential of solution spinning or hybrid methods which are attracting increased attention in the last decade is still to be discovered. The number of existing and emerging technologies can be optimized to obtain NFs of desired mechanical characteristics and drug loading/releasing properties. However, the wide choice of available polymers and therapeutic agents requires individual approach to their optimization, and the lack of comparative studies and detailed insight into the most recent methods, makes it difficult to single out the ideal or most appropriate production technique in regard to mechanical properties or loading/release of different therapeutic agents. Finally, the selection of the method of choice should also be dictated by their compatibility with the polymers and the therapeutic agent, administration mode as well as the interactions of the resulting NF with the biological system.

We have inserted this paragraph in Chapter 5 as our comment and summary on the current state of art on these issues. It is labelled in yellow.

4) The review is not focused on recent advancements, for instance, drug delivery based on core-sheath electrospun yarns, and its applications have to be included. 

‘Electrospun polymeric core-sheath yarns as drug eluting surgical sutures’

Response:

We thank the reviewer for pointing out this oversight. We included the suggested reference and some additional related ones in Section 4.1. Additions are marker in yellow.

Minor 

1) Line 14: Rewrite this sentence. Nanofibers are solid fibres that have a diameter under a micrometre. Replace ‘solid’ with polymeric/composite and ‘micrometre’ with nanoscale

Response:

We thank the Reviewer for the remark. We rewrote the sentence as instructed. The new sentence now reads: Nanofibers are polymeric/composite fibers which have a nanoscale diameter. It is labelled in yellow.

2) Elaborate the section 2.1, 2.2, 2.4 and 2.5

Response:

We thank the Reviewer for the suggestion. Since this review paper is oriented on recent developments in the application of nanofibers as the carriers for therapeutic molecules, we concentrated on more detailed description of methods that were used the most to prepare drug-loaded nanofibers in the last decade. We, therefore, elaborated Sections 2.5 (now 2.6) and 3D printing (now 2.4).

3) Section 3, titled ‘Drug loading and release’ should be changed to ‘Delivery of therapeutics’

Response:

Thank you for suggestion. The Section title was changed as requested.

4) Include recent citations. 

Response:

Thank you for remark. We added more recent citations throughout the text (38-41, 76-85, 88, 89, 130, 135-140). They can be seen in the labelled text and were additionally marked in yellow in References.

Round 2

Reviewer 2 Report

The authors have done an excellent job in revising the manuscript to address the comments.